# Divide-and-conquer: machine-learning integrates mammalian and viral traits with network features to predict virus-mammal associations

Maya Wardeh [1,2✉], Marcus S. C. Blagrove [3], Kieran J. Sharkey [2] & Matthew Baylis [1,4]

Our knowledge of viral host ranges remains limited. Completing this picture by identifying unknown hosts of known viruses is an important research aim that can help identify and mitigate zoonotic and animal-disease risks, such as spill-over from animal reservoirs into human populations. To address this knowledge-gap we apply a divide-and-conquer approach which separates viral, mammalian and network features into three unique perspectives, each predicting associations independently to enhance predictive power. Our approach predicts over 20,000 unknown associations between known viruses and susceptible mammalian species, suggesting that current knowledge underestimates the number of associations in wild and semi-domesticated mammals by a factor of 4.3, and the average potential mammalian host-range of viruses by a factor of 3.2. In particular, our results highlight a significant knowledge gap in the wild reservoirs of important zoonotic and domesticated mammals' viruses: specifically, lyssaviruses, bornaviruses and rotaviruses.

[1] Department of Livestock and One Health, Institute of Infection, Veterinary & Ecological Sciences, University of Liverpool, Liverpool, UK. [2] Department of Mathematical Sciences, University of Liverpool, Liverpool, UK. [3] Department of Evolution, Ecology and Behaviour, Institute of Infection, Veterinary & Ecological Sciences, University of Liverpool, Liverpool, UK. [4] Health Protection Research Unit in Emerging and Zoonotic Infections, University of Liverpool, Liverpool, UK. ✉email: maya.wardeh@liverpool.ac.uk

Thousands of viruses are known to affect mammals, with recent estimations indicating that less than 1% of mammalian viral diversity has been discovered to date[1]. Some of these viruses have a very narrow host range, whereas others such as rabies and West Nile viruses[2] have very wide host ranges (rabies can theoretically infect any mammal[3]). Host range is an important predictor of whether a virus is zoonotic[4] and therefore poses a risk to humans. For example, Severe acute respiratory syndrome-related (SARS-CoV) and Middle East respiratory syndrome-related (MERS-CoV) coronaviruses are both believed to have originated in bats, but through a host range that includes other mammals (e.g. palm civets[5], camels[6]) they have successfully infected humans. Most recently, SARS-CoV-2 has been found to have a relatively broad host range, including: bats; cats; ferrets; and a proposed intermediate host, Malayan pangolins, which may have facilitated spill-over to humans[7]. Knowing the potential host range of viruses is essential for efforts to mitigate the global burden of viral diseases[4,8].

However, our knowledge of the host range of viruses remains limited[1,4,9] and the information we have is hugely biased towards humans and domesticated mammals. For example, there is a significant gap between the number of known human viruses (274 species[10]), and those of wild primates (e.g. only 5 species in the toque macaque - *Macaca sinica*[10], and average of ~7 viruses per primate host[10]) which is largely a result of differential research effort. Surveillance and research efforts often intensify during and after significant outbreaks, leading to further biases; for instance, recent efforts to identify potential reservoirs of SARS-CoV-2 have led to the identification of two new virus species in wild pangolins (*Manis javanica* and *Manis pentadactyla*)[11], and a pangolin coronavirus[7], thereby doubling the number of known viruses of pangolins.

Despite these biases, the knowledge accumulated so far provides a valuable resource which can be exploited to estimate the extent to which we are under-observing associations between known viral agents and mammalian hosts. Networks, linking known viruses with their mammalian hosts, present a global view of sharing of these viruses amongst mammalian hosts. This sharing exhibits certain characteristics (e.g. DNA vs RNA viruses[12,13]; bats vs rodents[14]) which could only be captured at the global level. Various network topological features have been exploited to provide significant insight into patterns of pathogen sharing[14], disease emergence and spill-over events[15], and as means to predict missing links in a variety of host-pathogen networks[16,17] including helminths[18], and viruses[19,20].

Here, we express the topology of our virus-mammal network in terms of counts of potential motifs[21]. Motifs[22] are small subgraphs which constitute the building blocks of larger, more complex networks[23]. Motifs express specific functions or topological features of the underlying network, and have been used to capture complex and indirect interactions in a variety of systems including biology[24–26], ecology[27,28] and disease emergence[29]. We integrate this global view of viral sharing into a machine-learning driven framework to predict unknown (i.e. either potential or undocumented/unobserved) associations between known viruses and their mammalian hosts. The novelty of our framework lies in its multi-perspective approach whereby each possible virus-mammal association is predicted three times: 1) from the perspective of each of our mammals (e.g. based on the traits of the viruses known to infect wildcats - *Felis silvestris*, which other known viruses could also infect them?); 2) from the perspective of each of our viruses (e.g. based on the traits of mammalian species in which West Nile virus has been found to date, which other mammals can carry this virus?); and 3) from the perspective of the network linking known viruses with their mammalian hosts.

Our framework utilises 6,331 associations between 1896 viruses and 1436 terrestrial mammals, representing 0.23% of all possible associations between these mammals and viruses. It assesses how much these associations are underestimated by predicting which unknown species-level associations are likely to exist in nature (or do already exist but are yet undocumented). We aggregate these predictions to enhance estimation of the host-range of known mammalian viruses, and to highlight variation in the degree of underestimation at the level of mammalian order (particularly in wild and semi-domesticated species), and viral group (Baltimore classification), family, and genus. In addition, we highlight knowledge gaps in mammalian species susceptible to known zoonoses and equivalent viruses in important domesticated mammals. By investigating this underestimation from three separate points of view, we enhance the overall predictive performance and capture local (at the level of a single viral or mammalian species), as well as global (aggregated) variations in our knowledge gaps.

## Results

Our framework to predict unknown associations between known viruses and potential mammalian hosts or susceptible species comprised three distinct perspectives: viral, mammalian and network. Each perspective produced predictions from a unique vantage point (that of each virus, each mammal, and the network connecting them respectively). Subsequently, their results were consolidated via majority voting. This approach suggested that 20,832 (median, 90% CI = [2,736, 97,062], hereafter values in square brackets represent 90% CI) unknown associations potentially exist between our mammals and their known viruses, (18,920 [2,440, 91,517] in wild or semi-domesticated mammals). Number of unknown associations predicted by each perspective individually were as follows: mammalian only = 41,537 [4,275, 23,8971], viral only = 21,352 [2,536, 95,630], and network only = 76,081 [27,738, 20,5814]. Our results indicated a ~4.29-fold increase ([~1.43, ~16.33]) in virus-mammal associations (~4.89 [~1.5, ~19.81] in wild and semi-domesticated mammals).

Additionally, we trained an independent pipeline including only the 3534 supported by evidence extracted from meta-data accompanying nucleotide sequences, as indexed in EID2 (55.82% of all associations - see Methods section and Supplementary Results 8). Our sequence-evidence pipeline indicated that 15,721 (median, 90% CI = [1,603, 88,553]) unknown associations could potentially exist (13,930 [1,298, 83,043] in wild or semi-domesticated mammals).

In the following subsections we first illustrate the mechanism of our framework via an example, then further explore the predictive power of our approach for viruses and mammals.

**Example**. Our multi-perspective framework generates predictions for each known or unknown virus-mammal association (2,722,656 possible associations between 1,896 viruses and 1,436 terrestrial mammals). We highlight this functionality using two examples (Fig. 1). West Nile virus (WNV) a flavivirus with wide host range, and the bat *Rousettus leschenaultia* (order: Chiroptera). We first consider each of our perspectives separately, and then showcase how these perspectives are consolidated to produce final predictions.

1) The mammalian perspective: our mammalian perspective models, trained with features expressing viral traits (Table 1), suggested a median of 90 [17, 410] unknown associations between WNV and terrestrial mammals could form when predicting virus-mammal associations based on viral features alone – a ~2.61-fold increase [~1.3, ~8.32]. Similarly, our results indicated that 64 [4, 331] new associations could form between

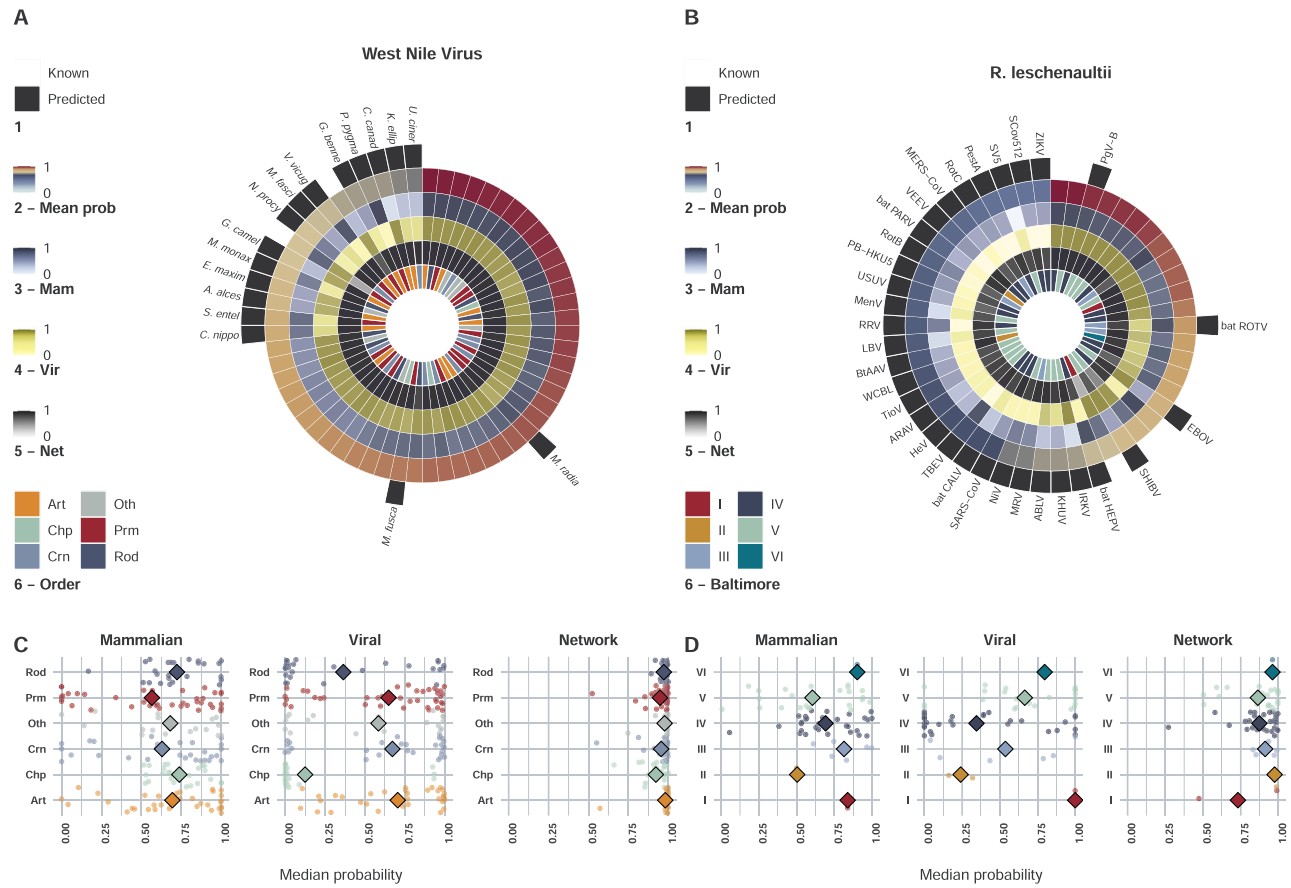

**Fig. 1 Example showcasing final and intermediate predictions of West Nile Virus (WNV), and *Rousettus leschenaultii*.** Panel **A** Top 60 predicted mammalian species susceptible to WNV. Mammals were ordered by mean probability of predictions derived from mammalian (all models), viral (WNV models) and network perspectives, and top 60 were selected. Circles represent the following information in order: 1) whether the association is known (documented in our sources) or not (potential or undocumented). Hosts are omitted for known associations. 2) Mean probability of the three perspectives (per association). 3) Median mammalian perspective probabilities of predicted associations. These probabilities are obtained from 3000 models (50 replicate models for each mammal), trained with viral features – SMOTE class balancing. 4) Median viral perspective probabilities of predicted associations (50 WNV replicate models trained with mammalian features – SMOTE class balancing). 5) Median network perspective probabilities of predicted associations (100 replicate models, balanced under-sampling). 6) Taxonomic order of predicted susceptible species. Orders are shortened as follows: Artiodactyla (Art), Carnivora (Crn), Chiroptera (Chp), primates (Prm), Rodentia (Rod), and Others (Oth). Panel **B** Top 50 predicted viruses of *R. leschenaultii*. Viruses were ordered by mean probability of predictions derived from mammalian (*R. leschenaultii* models), viral (all models) and network perspectives. Circles as per Panel **A**. Baltimore represents Baltimore classification. Panel **C** Median probability of predicted WNV-mammal associations in each of the three perspectives per mammalian order. Points represent susceptible species predicted by voting (at least two of the three perspectives – $n =$ 137). Median ensemble probability is computed in each perspective (50 replicate models for each virus/mammal, 100 replicate network models). Predictions derived from each perspective at 0.5 probability cut-off. Supplementary Data 1 presents full WNV results. Panel **D** Median probability of virus-*R. leschenaultii* associations in the three perspectives per Baltimore group. Points represent susceptible species predicted by voting (at least two of the three perspectives – $n = 64$), predictions are derived as per panel **C**. Supplementary Data 2 lists full results for R. leschenaultii. Supplementary Fig. 7 illustrate the results when research effort into viruses and mammals is included in mammalian and viral perspectives, respectively.

our selected mammal (*R. leschenaultia*) and our viruses – a ~4.37-fold increase [~1.21, ~18.42] (Supplementary Results 4).

(2) The viral perspective: our viral models, trained with features expressing mammalian traits (Table 2), indicated a median of 48 [0, 214] new hosts of WNV (~1.86- fold increase [~1, 4.82]). Results for our example mammal (*R. leschenaultia*) suggested 18 [3, 76], existing viruses could be found in this host (~1.95-fold increase [~1.16, ~5.00]) - Supplementary Results 5).

(3) The network perspective: Our network models indicated a median of 721 [448, 1,317] (~13.88 [9, 24.52] fold increase) unknown associations between WNV and terrestrial mammals, and that 246 [91, 336] existing viruses could be found in our selected host (*R. leschenaultia*), equivalent to a ~13.95 [5.79, ~18.68] fold increase (Supplementary Results 6).

Considering that each of the above perspectives approached the problem of predicting virus-mammal associations from a different angle, the agreement between these perspectives varied. In the case of WNV: mammalian and viral perspectives achieved 92.3% agreement [72.6%–98.5%]; mammals and network perspectives had 55.3% agreement [33.4%–69.5%]; and viruses and network had 52.9% agreement [19.8%–68.7%]. In the case of *R. leschenaultia* these numbers were as follows: 96.15% [82.44%, 99.58%], 87.24% [76.37%, 95.04%], and 87.61% [75.90%, 95.25%], respectively. The agreements between our perspectives across the 2,722,656 possible associations were as follows: 98.04% [90.36%, 99.73%] between mammalian and viral perspectives, 96.71% [88.62%, 98.92%] between mammalian and network perspectives, and 97.11% [91.57%, 98.95%] between viral and network perspectives.

**Table 1 Viral traits & features used to build our mammalian models.**

| Category | Viral Feature | Data type | Reason for inclusion |
|---|---|---|---|
| Host-driven | Mean phylogenetic distance between hosts | Continuous | Capturing phylogenetic and ecological distances between each virus' known hosts and each mammal in our study. |
| | Mean ecological distance between hosts | | |
| | Maximum phylogenetic breadth[4] | | Greater phylogenetic breadth indicates more generalist potential of the virus. |
| Virus genome & capsid | RNA | Binary | RNA viruses mutate/adapt faster[65], and are generally deactivate quickly when exposed to the environment. |
| | Retro-transcribing | | Retroviruses are generally very conserved[66], have to enter the nucleus[67] and insert into the genome. Additional steps may require specificity and limit range. |
| | Negative sense/positive sense | | Sense affects replication cycle and range of host enzymes needed. |
| | Circular/linear | | Circular/linear genome affects enlisting host enzymes for replication and translation[68]. |
| | Monopartite/segmented | | Segmented viruses can undergo recombination if two strains of the same virus infect a cell[69]. This can lead to host range changes of segments of the genome. |
| | Enveloped | | Envelopes are derived from the host cell membrane, so can affect specific-host immune activation. Enveloped viruses deactivate rapidly in the external environment (often requiring direct transfer). The envelope will change upon infection of a new host[70]. |
| | GC-content | Continuous | High GC content usually leads to higher thermo-stability of the genome[71]. |
| | Genome size | | Genome size is indicative of many aspects of the virus such as complexity, DNA/RNA, and replication type. |
| Virus replication, release, and cell entry | Cytoplasm | Binary | Replication site is linked to RNA/DNA genome – if a virus has a DNA stage it must replicate in the nucleus and overcome additional cell barriers. |
| | Release | Categorical | Affects rate of virus production, cell life-span and means of presentation to the immune system[72]. |
| | Cell entry | | Availability of receptors influences potential host range. |
| Transmission routes | 8 main transmission routes | Binary for each route | Route(s) of transmission affected by structure/stability of virus and nature of interaction between potential hosts. |

We trained a suite of models for each mammalian species with two or more known viruses (*n* = 699). Each model comprised the below described features (response variable = 1 if the virus is known to associate with the focal mammalian species, 0 otherwise – methods section provides further details). Full description of these features, their sources and justification are listed in Supplementary Note 2.

After voting, our framework suggested that a median of 117 [15, 509] new or undetected associations could be missing between WNV and terrestrial mammals (~3.45-fold increase [~1.3, ~12.2]). Similarly, our results indicated that *R. leschenaultia* could be susceptible to an additional 45 [5, 235] viruses that were not captured in our input (~1.37-fold increase [~1.26, ~13.37]). Figure 1 illustrates top predicted and detected associations for WNV (Supplementary Data 1) and *R. leschenaultia* (Supplementary Data 2). Supplementary Results 1 illustrate results with research effort into viruses, and mammals included as a predictor in our mammalian and viral perspective models, respectively. Predictions with and without research effort incorporated into models trained in these perspectives broadly agreed.

**Relative importance of viral features**. Our multi-perspective approach trained a suite of models for each mammalian species with two or more known viruses (*n* = 699, response variable = 1 if the virus is known to associate with the focal mammalian species, 0 otherwise). This enabled us to assess the relative importance (influence) of viral traits (Table 1) to each of our mammalian models. This in turn showcased variations of how these viral traits contribute to the models at the level of individual species (e.g. humans), and at an aggregated level (e.g. by order or domestication status). The results, highlighted in Fig. 2A, indicate that mean phylogenetic (median = 95.4% [75.6%, 100%]) and

mean ecological (90.90% [43.50%, 100%]) distances between potential and known hosts of each virus were the top predictors of associations between the focal host and each of the input viruses. Maximum phylogenetic breadth was also important (74.70%, [16.60%, 100%]).

**Mammalian host range**. Our results suggested that the average mammalian host range of our viruses is 14.33 [4.78, 54.53] (average fold increase of ~3.18 [~1.23, ~9.86] in number of hosts detected per virus). Overall, RNA viruses had the average host range of 21.65 [7.01, 82.96] hosts (~4.00- fold increase [~1.34, ~14.15]). DNA viruses, on the other hand, had 7.85 [2.81, 29.47] hosts on average (~2.43 [~1.14, ~6.89] fold increase). Table 3 lists the results of our framework at Baltimore group level and selected family and transmission routes of our viruses. Figure 2 illustrates predicted mammalian host range of our viruses (Fig. 2B, Supplementary Data 3), and the increase in predicted number of viruses per species in species-rich mammalian orders of interest (Fig. 2C, Supplementary Data 4).

**Relative importance of mammalian features**. We trained a suite of models for each virus species with two or more known mammalian hosts (*n* = 556, response variable = 1 if the mammal is known to associate with the focal virus species, 0 otherwise). This allowed us to calculate relative importance of mammalian

**Table 2 mammalian traits & features used to build our viral models.**

| Category | Mammalian feature | Reason for inclusion |
|---|---|---|
| Phylogeny | Mean phylogenetic distance to known hosts. | Linked to sharing of viruses between mammals[4,73,74]. |
|  | Evolutionary distinctiveness | Can correlate negatively with pathogen species richness[75]. |
| Taxonomy & domestication | Order & family | Can affect host-pathogen[76], particularly viral, associations[4]. |
|  | Domestication | Might influence sharing of viruses between host groups. Domesticated mammals and human might share more viruses with each other than related wild species. |
| Ecological traits | Morphological traits (body mass) | A key feature in terms of metabolism and adaption to environment. |
|  | Life-history traits (Maximum age, age at sexual maturity, activity cycle, and migration) Reproductive traits (gestation period length, litters per year, litter size and weaning age) | Potentially relevant in terms of within-host dynamics of viruses. |
|  | Habitat utilisation | Similar habitat utilisation might correlate with contact with similar viruses. |
|  | Diet (proportional use of 10 categories) | Similar dietary habit might associate with similar viral assemblage. |
|  | Mean ecological distance | Indicates if a potential host species is *ecologically* close to or distant from the virus' preferred host range. We based this distance on a generalised form of Gower's distance matrices[77,78] incorporating all ecological traits. |
| Geo-spatial features | Geographical range (area size) | Might lead to exposure to larger number or more diverse viruses. |
|  | Climate (mean temperature & precipitation) | Climate has been shown to influence a number of human and domestic mammal pathogens[79,80]. |
|  | Natural land cover diversity/Agriculture and farming diversity | These factors have been found to influence certain categories of host-pathogen associations[81]. Supplementary Note 3 lists further details of mammalian geo-spatial feature extraction. |
|  | Mammalian biodiversity |  |
|  | Urbanisation/human population |  |

We trained a suite of models for each virus species with two or more known mammalian hosts ($n = 556$). Each model comprised the below described features (response variable $= 1$ if the mammal is known to associate with the focal virus species, 0 otherwise – methods section provides further details). Full description of these features, their sources and justification are listed in Supplementary Note 3.

traits (Table 2) to our viral models. We were also able to capture variations in how these features contribute to our viral models at various levels (e.g. Baltimore classification, or transmission route) as highlighted in Fig. 3A. Our results indicated that distances to known hosts of viruses were the top predictor of associations between the focal virus and our terrestrial mammals. The breakdown was: 1) mean phylogenetic distance - all viruses = 98.75% [93.01%, 100%], DNA = 99.48% [96.03%, 100%], RNA = [91.93%, 100%]; 2) mean ecological distance all viruses = 94.39% [71.86%, 100%], DNA = 96.36% [80.99%, 100%], RNA = [69.48%, 100%]. In addition, life-history traits significantly improved our models, in particular: longevity (all viruses = 60.9% [12.12%, 98.88%], DNA = 68.03% [11.22%, 99.69%], RNA = [13.55%, 96.37%]); body mass (all viruses = 62.92% [5.4%, 97.65%], DNA = 72.75% [18.49%, 100%], RNA = 57.45% [4.32%, 95.5%]); and reproductive traits (all viruses = 53.37% [5.67%, 95.99%]%, DNA = 59.46% [8.27%, 99.32%], RNA = 50.17% [4.85%, 92.17%]).

**Wild and semi-domesticated susceptible mammalian hosts of viruses.** our framework indicated ~4.28 -fold increase [~1.2, ~14.64] of the number of virus species in wild and or semi-domesticated mammalian hosts (16.86 [4.95, 68.5] viruses on average per mammalian species). These results indicated an average of 13.45 [1.73, 65.04] unobserved virus species for each wild or semi-domesticated mammalian host (known viruses that are yet to be associated with these mammals). Our framework highlighted differences in the number of viruses predicted per order (Table 4). Figure 3 illustrates the predicted number of viruses in wild or semi-domesticated mammal by mammalian host range (Fig. 3B, Supplementary Data 5), and the top 18 virus genera (per number of host-virus associations) in selected orders (Fig. 3C, Supplementary Data 6). Supplementary Results 1 lists the results with the inclusion of research effort into mammalian species in our viral perspective models.

**Network perspective - *Potential motifs*.** We quantified the topology of the network linking virus and mammal species by means of counts of potential motifs[21]. Figure 4 illustrates how potential motifs are captured in our network. Briefly, for each virus-mammal association for which we want to make predictions ($n = 2,722,656$, of which 6,331 are supported by our evidence, see methods section), we "force insert" this focal association into our network (Fig. 4A, B) and enumerate all instances of 3 ($n = 2$), 4 ($n = 6$), and 5-node ($n = 20$) potential motifs in which this association might feature if it actually existed[21] (Fig. 4C visualises these different motifs). Following this process, a features-set is generated comprising the counts potential motifs for all included associations. Figure 4D illustrates the count of motifs (logged) grouped by mammalian order and virus Baltimore classification.

**Relative importance of network (motif) features.** Figure 4E illustrates that M4.1 was the most important feature in our network models: median = 100% [90.19%, 100%]. Followed by: M5.1 = 97.84% [89.19%, 99.93%], M5.7 = 98.8 97.22% [87.7%, 98.77%] and M4.6 = 96.75% [86.13%, 100%]. Research effort of viruses and mammals had relative importance = 90.26% [82.94%, 95.36%], 88.42% [78.38%, 94.87%] respectively. Overall, 5-node motif-features had median relative influence = 75.06% [1.21%, 98.14%]; whereas 3 and 4-node motif-features had relative influence = 71.69% [55.76%, 85.34%], and 61.06% [27.14%, 100%], respectively. Supplementary Fig. 29 illustrate the partial dependence of network perspective models on each of our network features.

**Validation.** We validated our framework in three ways: 1) against a held-out test set; 2) by systematically removing selected known viral-mammalian associations and attempting to predict them; and 3) against external data source, comprising viral-mammalian associations extracted using an exhaustive literature search targeting wild mammals and their viruses[4,30].

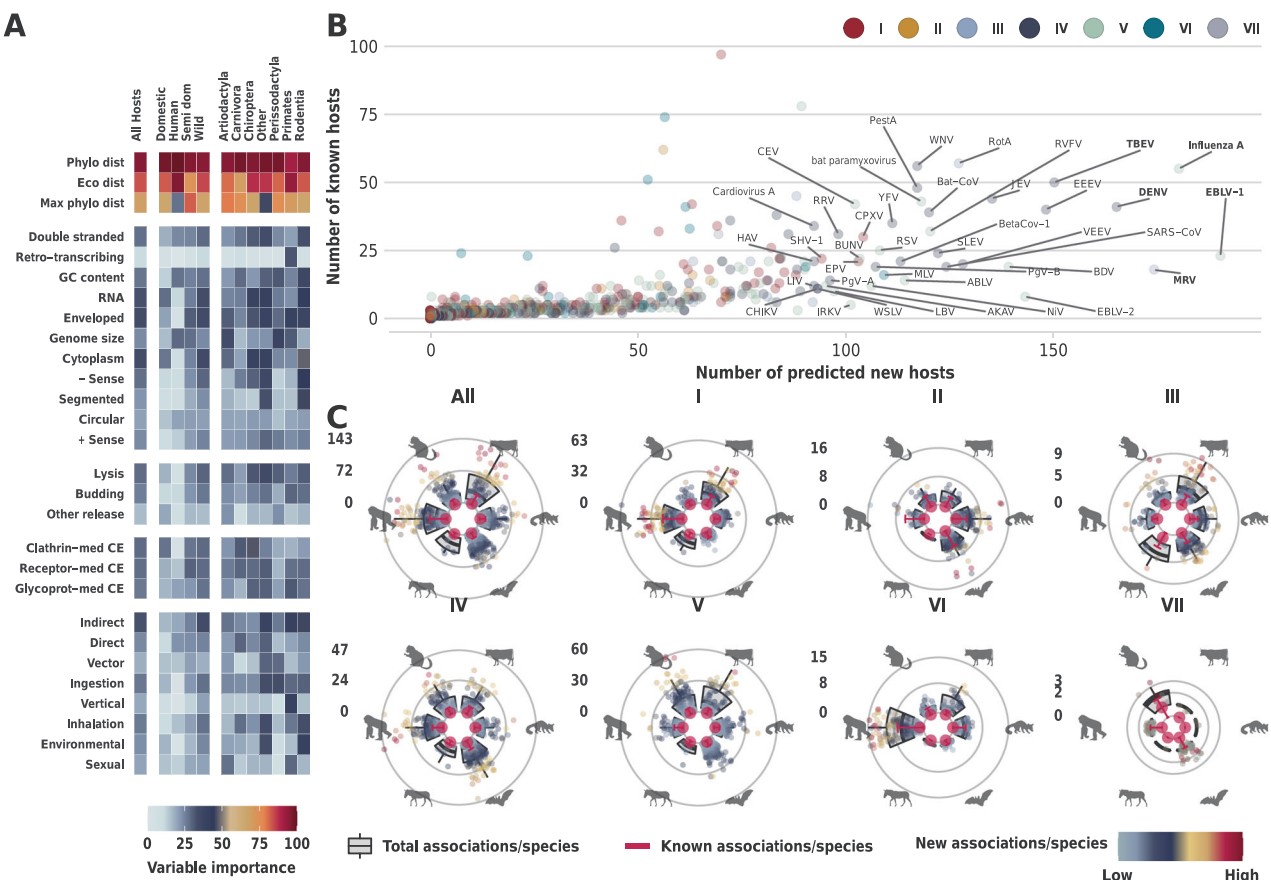

**Fig. 2 Results (viruses).** Panel **A** Variable importance (relative contribution) of viral traits to mammalian perspective models. Variable importance is calculated for each constituent ensemble ($n = 699$) of our mammalian perspective (median of a suite of 50 replicate models, trained with viral features, with SMOTE sampling), and then aggregated (mean) per each reported group (columns). Panel B – Number of known and new mammalian species associated with each virus. Rabies lyssavirus was excluded from panel **B** to allow for better visualisation. Top 40 (by number of new hosts) are labelled. Species in bold have over 150 predicted hosts (Supplementary Data 3 lists details of these viruses including CI). Panel **C** Predicted number of viruses per species of wild and semi-domesticated mammals (group by mammalian order). Following orders (clockwise) are presented: *Artiodactyla*, *Carnivora*, *Chiroptera*, *Perissodactyla*, *Primates*, and *Rodentia*. Source of the silhouette graphics is PhyloPic.org. (Supplementary Data 4 lists aggregated results per mammalian order). Circles represent each mammalian species (with predicted viruses > 0), coloured by number of known viruses previously not associated with this species. Boxplots indicate median (centre), the 25th and 75th percentiles (bounds of box) and inter quantile range (whiskers) and are aggregated at the order level. Large red circles with error bars (90% CI) illustrate the median number of known viruses per species in each order. Number of species presented ($n$) is as follows: All = 1293 (*Artiodactyla* = 104, *Carnivora* = 177, *Chiroptera* = 548, *Perissodactyla* = 11, *Primates* = 171, and *Rodentia* = 282); Group I = 666 (94, 109, 156, 10, 160, 137); Group II = 371 (32, 120, 111, 1, 54, 53); Group III = 410 (87,62,123,9,51,78); Group IV = 739 (98, 102, 221, 9, 148, 161); Group V = 1129 (87, 173, 528, 8, 107, 226); Group VI = 358 (55, 64, 30, 6, 139, 64); and Group VII = 110 (3,2,53,1,43,8). Supplementary Fig. 8 presents results derived with research effort into mammalian hosts and viruses included in the constituent models trained in the viral and mammalian perspectives, respectively.

Our held-out test set comprised 15% of all data (randomly selected, $n = 407,265$; 954 known virus-mammal associations, see methods below). We removed this set from our network, computed network features (motifs), and trained constituent models in each perspective with the remainder data. We then estimated our framework performance metrics against the held-out test set. Our framework achieved overall AUC = 0.938 [0.862–0.959], F1-Score = 0.284 [0.464–0.124], and TSS = 0.876 [0.724–0.918], when trained without including research effort in its mammalian and viral perspectives. When research effort was included in these perspectives, performance metrics were as follows: AUC = 0.920 [0.823, 0.944], F1-Score = 0.272 [0.526, 0.093], and TSS = 0.840 [0.646, 0.888].

The performance of our voting approach was better than any individual perspective, or combination of perspectives (Supplementary Tables 8–11). The most significant improvement was in F1-score, where individual perspectives scores were as follows: network = 0.104 [0.210–0.051], mammalian = 0.115 [0.009–0.064] (0.131 [0.284–0.035] with research effort), and viral = 0.181 [0.374–0.074] (0.196 [0.373–0.067]).

Additionally, we conducted a systematic test to predict removed virus-mammal associations. In this test, we systematically removed one known virus-mammal association at a time from our framework, recalculated all inputs (including from network) and attempted to predict these removed associations. Our framework succeeded in predicting 90% of removed associations (90.70% for associations removed for viruses, 89.92% for associations removed from mammals, Supplementary Results 3).

Finally, our framework predicted 84.02% [77.69%, 89.60%] of the externally obtained viral-mammalian associations (with detection quality > 0) where both host and virus were included in our pipeline, and 77.82% [68.46%, 86.51%] (any detection quality). When including research effort in our mammalian and viral perspectives, these results were: 84.47% [78.15%, 89.60%], and 78.41% [68.83%, 86.37%], respectively.

**Table 3 Predicted range of susceptible mammalian species of viruses per Baltimore group, family (top 15 families, ranked by fold increase) and transmission route.**

| Baltimore classification | Predicted range (~fold increase) | Family | | Predicted range (~fold increase) |
|---|---|---|---|---|
| Group I (dsDNA) | 8.63 [3, 30.43] (~2.59 [-1.16, -6.94]) | Bornaviridae | V | 71 [15.5, 293.25] (~9.51 [-2.08, -42.22]) |
| Group II (ssDNA) | 5.47 [2.19, 24.88] (~2.04 [-1.07, -6.56]) | Orthomyxoviridae | V | 60.25 [15, 196.5] (~7.76 [-1.62, -27.19]) |
| Group III (dsRNA) | 27.15 [7.96, 93.11] (~4.04 [-1.41, -11.94]) | Rhabdoviridae | V | 52.8 [23.68, 149.09] (~7.33 [-1.81, -24.03]) |
| Group IV ((+)ssRNA) | 17.64 [5.34, 65.29] (~3.49 [-1.26, -10.9]) | Hepeviridae | IV | 70.67 [25.33, 220] (~6.67 [-2.49, -15.54]) |
| Group V ((−)ssRNA) | 24.91 [8.36, 100.53] (~4.44 [-1.39, -18.08]) | Filoviridae | V | 31.75 [7, 155.62] (~3.77 [-1.3, -25.37]) |
| Group VI (ssRNA-RT) | 26.68 [10.26, 94.58] (~4.99 [-1.54, -15.36]) | Togaviridae | IV | 48.5 [12.45, 161.65] (~5.71 [-1.52, -16.95]) |
| Group V (dsDNA-RT) | 19.29 [7.29, 109.43] (~2.53 [-1.35, -14.55]) | Flaviviridae | IV | 40.59 [11.26, 131.77] (~5.09 [-1.37, -16.14]) |
| | | Retroviridae | VI | 26.68 [10.26, 94.58] (~4.99 [-1.54, -15.36]) |
| Transmission route | | Coronaviridae | IV | 22.86 [6.23, 94.89] (~4.81 [-1.44, -17.85]) |
| Direct | 14.67 [5.07, 55] (~3.29 [-1.25, -10.49]) | Poxviridae | I | 23.22 [7.76, 88.76] (~4.77 [-1.39, -15.74]) |
| Direct sexual | 18.26 [6.05, 60.05] (~3.18 [-1.27, -9.17]) | Reoviridae | III | 32.5 [9.39, 111.21] (~4.56 [-1.49, -13.71]) |
| Direct vertical | 20.26 [6.81, 68.79] (~3.44 [-1.31, -10.48]) | Paramyxoviridae | V | 26.28 [9.39, 98.79] (~4.46 [-1.61, -14.4]) |
| Indirect | 20.02 [7.35, 71.38] (~3.41 [-1.27, -11.47]) | Phenuiviridae | V | 26.94 [6.35, 124.18] (~4.25 [-1.23, -20.34]) |
| Ingestion | 10.7 [4.11, 39.46] (~2.52 [-1.15, -7.69]) | Peribunyaviridae | V | 20.09 [5.85, 90.45] (~4.15 [-1.39, -19.03]) |
| Inhalation | 14.53 [4.6, 59.87] (~3.29 [-1.24, -11.73]) | Hantaviridae | V | 15.61 [4.83, 77.59] (~3.61 [-1.23, -17.13]) |
| Environmental | 20.32 [6.25, 82.58] (~3.81 [-1.29, -14.12]) | Picornaviridae | IV | 13.62 [4.45, 52.93] (~3.55 [-1.32, -10.7]) |
| Vector | 30.1 [8.38, 117.44] (~4.73 [-1.42, -18.26]) | Pneumoviridae | V | 28.89 [8.67, 107.56] (~3.47 [-1.18, -12.78]) |

Results with research effort included into our mammalian and viral perspective models are reported in Supplementary Table 6.

## Discussion

Overall, we predict a 5.35-fold increase in associations between wild and semi-domesticated mammalian hosts and known zoonotic viruses (found in humans, excluding rabies virus). Similarly, our results indicate a 5.20-fold increase between wild and semi-domesticated mammals and viruses of economically important domestic species (e.g. livestock and pets). Bats and rodents, which have been associated with recent outbreaks of emerging viruses such as coronaviruses[31] and hantaviruses[32], are linked with increased risk of zoonotic viruses[4,13,30,33]. Our results could potentially enable targeted surveillance of rodents and bats for known viruses not yet associated with species in these orders: we predict a 5.55-fold (2.69 per species) and a 5.45-fold (3.77) increases respectively (Fig. 2C, Supplementary Data 6). The fold increases are higher for zoonotic viruses and viruses observed in economically important domestic species, where for bats we predict a 7.42-fold (2.30 per species) and an 8.29-fold increase (2.42 per species) respectively. Whereas for rodents we predict a 6.43-fold (3.69) and a 7.7-fold increase (2.92), respectively.

The increase in associations indicates a knowledge-gap across mammalian species that are potentially susceptible to these viruses. For bats the largest fold increase was in group III viruses with an 8.72-fold-increase (1.43 per species, group IV had the highest fold increase per species, 2.26), whereas in rodents the highest increase was in group V viruses - a 6.23-fold-increase (3.49 per species).

The largest significant fold increases in included bats were with the group V Lyssaviruses (excluding rabies), a family of viruses causing an array of medically and veterinary important rabies-like diseases in a wide range of mammals[34,35], with a 10.4-fold increase in the number of predicted associations (Fig. 3C, Supplementary Data 6). Group V Bornaviruses, which cause a range of encephalitic diseases in mammals including the fatal Borna disease[36] (sad horse disease) common in horses and other domesticated animals, had a 23 and a 12-fold increase in associations in bats and rodents, respectively. Finally, group III Rotaviruses had an 8.11-fold increase in bats – rotaviruses are the most common cause of diarrhoeal diseases in children and are of particular concern in developing countries[37].

Analogous to bats and rodents being important hosts of zoonotic viruses, wild ruminants are key in the maintenance and circulation of viruses affecting ruminant livestock[38]. Our framework highlights this knowledge-gap by predicting a 7.77-fold increase in number of associations between wild and semi-domesticated ruminants and known viruses (3.37-fold increase per species, Fig. 2C, Supplementary Table 14); and a 10.11-fold increase in associations between these ruminants and observed zoonotic viruses (2.25-fold per species). Furthermore, our model predicted a significant increase in the mammalian host range of important livestock viruses including: a 7.45-fold increase in range of Venezuelan equine encephalitis virus (Group IV, Togaviridae); a 5.33-fold increase in range of Schmallenberg orthobunyavirus (Group V, Peribunyaviridae); and a 2.96-fold increase in range of bluetongue virus (Group III, Reoviridae).

These results demonstrate that our approach can highlight large numbers of potentially missing associations of medically- and veterinary-important viruses and their potential hosts. For instance, we predicted 13 genera of viruses in three species of *lynx* (*lynx canadensis*, *lynx rufus* and *lynx pardinus*) which were not associated with the lynx in our input data, including Nipah virus. Such information can be used to better understand the risk to people and livestock from these hosts. There are several reasons for which virus-mammal associations may have been disproportionately under-described, which can be categorised as follows:

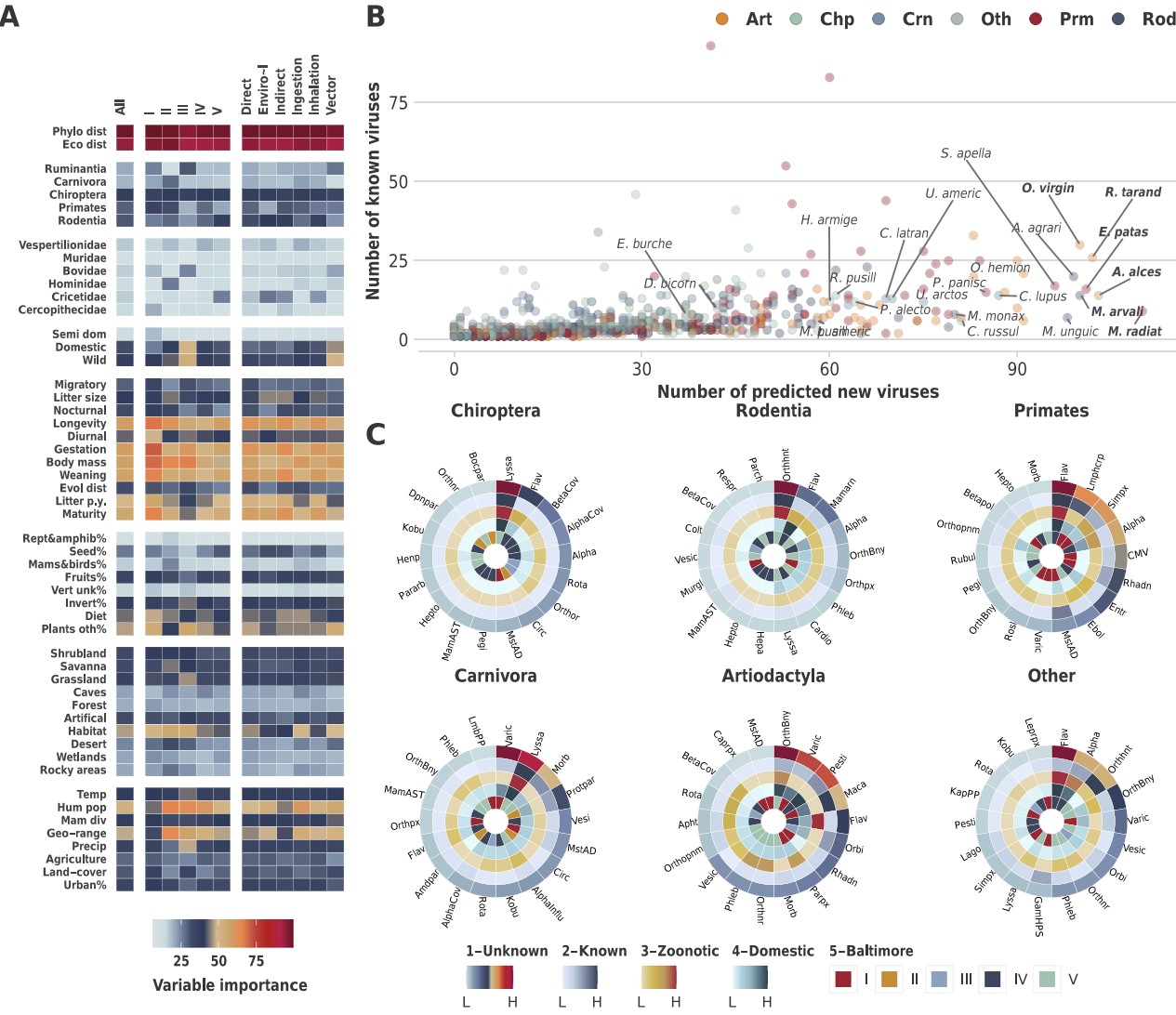

**Fig. 3 Results (Mammals).** Panel **A** Variable importance (relative contribution) of mammalian traits to viral perspective models. Variable importance is calculated for each constituent model ($n = 556$) of our viral perspective (trained with mammalian features), and then aggregated (median) per each reported group (columns). Panel **B** Number of known and new viruses associated with each mammal. Labelled mammals are as follows: top 4 (by number of new viruses) for each of Artiodactyla, Carnivora, Chiroptera, Primates, Rodentia, and other orders. Species in bold have 100 or more predicted viruses (Supplementary Data 5). Panel **C** Top 18 genera (by number of predicted wild or semi-domesticated mammalian host species) in selected orders (Other indicated results for all orders not included in the first five circles). Each order figure comprises the following circles (from outside to inside): 1) Number of hosts predicted to have an association with viruses within the viral genus. 2) Number of hosts detected to have association. 3) Number of hosts predicted to harbour viral zoonoses (i.e. known or predicted to share at least one virus species with humans). 4) Number of hosts predicted to share viruses with domesticated mammals of economic significance (domesticated mammals in orders: Artiodactyla, Carnivora, Lagomorpha and Perissodactyla). 5) Baltimore classification of the selected genera (Supplementary Data 6). Supplementary Fig. 9 presents results derived with research effort into mammalian hosts and viruses included in the constituent models trained in the viral and mammalian perspectives, respectively.

1. Public health, food security and economically driven research biases: Most of our current knowledge of infectious agents, including viruses, is centred upon humans. Second to humans (37.1% of captured mammalian research effort), agricultural and companion animals tend to receive significantly more research effort (~15% of captured mammalian research effort). Examples include the well-studied microbiome of domestic cats (*Felis catus*, 57 known virus species) compared with the understudied microbiome of wild felines (e.g. *Felis silvestris*, 13 known viruses – these expanded to 51 using our framework). Linked to this is wealthier countries producing a larger research volume, and hence interactions common within or of importance to such countries are more likely to be described.

2. Practical limitations: infectious agents of endangered and rare mammalian species, and mammalian species found predominantly in remote regions, we suspect, are less likely to be characterised due to difficulties in sampling these mammals in their natural habitats. The same likely applies to viruses that are less common in mammals (e.g. avian pathogens). Nevertheless, our approach can capture and expand associations of both rare viruses (found in one or two species), and understudied mammalian species, due to separation of perspectives. If a virus is rare, our approach would capture potentially susceptible mammals via the network and mammalian perspectives. Similarly, if a mammalian species is rarely studied, then we would still capture viruses potentially found in this mammalian species

**Table 4 Predicted number of viruses per top 15 orders by fold increase in number of viruses predicted in wild or semi-domesticated mammalian hosts (per species).**

| Order/sub-order | Included species | Fold increase/species | Virus range/species | New viruses/species |
|---|---|---|---|---|
| *Tylopoda (part Artiodactyla)* | 6 | ~12 [-1.75, -30.45] | 31.5 [4, 125] | 29 [1.5, 122.5] |
| *Ruminantia (part Artiodactyla)* | 99 | ~9.12 [-1.69, -23.53] | 41.45 [10.92, 126.55] | 36 [5.61, 121.07] |
| *Primates* | 172 | ~7.12 [-1.48, -20.88] | 34.46 [10.24, 114.62] | 27.77 [3.86, 107.89] |
| *Suina (part Artiodactyla)* | 13 | ~7.12 [-0.88, -28.01] | 21.83 [3.33, 112.83] | 18.92 [1.33, 109.83] |
| *Perissodactyla* | 14 | ~5.74 [-1.67, -17.78] | 25.18 [8, 97.09] | 20.64 [3.73, 92.55] |
| *Cingulata* | 1 | ~5.67 [-1.67, -27] | 17 [5, 107] | 14 [2, 104] |
| *Lagomorpha* | 13 | ~5.27 [-1.9, -16.43] | 18 [5.25, 74.83] | 15.08 [2.42, 71.92] |
| *Rodentia* | 287 | ~4.79 [-1.18, -18.72] | 15.22 [3.54, 74.84] | 12.65 [1.26, 72.22] |
| *Carnivora* | 180 | ~3.91 [-1.23, -14.84] | 18.3 [5.99, 78.38] | 14.11 [1.94, 74.15] |
| *Hippopotamidae (part Artiodactyla)* | 2 | ~3.00 [-1.00, -10.5] | 3 [1, 20] | 2 [0, 19] |
| *Chiroptera* | 548 | ~2.79 [-1.08, -9.5] | 9.51 [3.11, 41.66] | 7.11 [0.84, 39.24] |
| *Scandentia* | 3 | ~2.15 [-0.73, -10.34] | 11 [4.33, 55.33] | 5.67 [1, 50] |
| *Didelphimorphia* | 5 | ~2.13 [-0.93, -8.57] | 4.6 [1.6, 27] | 3 [0, 25.2] |
| *Eulipotyphla* | 44 | ~2.04 [-0.86, -13.78] | 7.14 [2.45, 55] | 4.73 [0.36, 52.45] |
| *Pilosa* | 5 | ~1.84 [-0.6, -17.98] | 7.8 [3.8, 68.4] | 3.4 [0, 63.8] |

Results are ordered by descending fold increase. Values are derived per species and averaged per order. Results with research effort included into our mammalian and viral perspective models are reported in Supplementary Table 7.

via the network and viral perspectives. Overall, our voting framework was able to expand the host range of rare viruses (known hosts ≤ 2, $n = 1450$) from 1,619 to 4,174 (~2.16 average increase per rare virus). Virus range of rare mammals (known viruses ≤ 2, $n = 954$) was also increased from 1150 to 4318 (~3.21 average fold increase per mammal).

3. Biological reasons: virus-mammal associations which produce more visible or marked effects are more likely to have been studied[39]. For instance, fertility or physically observable interactions are more likely to be over-studied, whilst potentially important asymptomatic interactions, or interactions where a cross-immunity from related viruses masks observable symptoms, may potentially remain unnoticed and hence understudied. Furthermore, co-evolution between virus and primary host often results in a less severe phenotype[40], whilst the same virus in an incidental host may result in more marked and hence more studied disease. Examples include Ebola viruses presenting minimal symptoms in bats but severe disease with high mortality in humans[41]; analogous interactions where the former host may have been unobserved are likely to be plentiful. For example, our framework indicated that 34 species of bats could be susceptible Ebolaviruses. Recently, advances in metagenomics have enhanced viral discovery in hosts, enabling cheap and rapid identification and sequencing of host viromes. This approach mitigates many historical 'top-down' limitations mentioned above, enabling simple identification of e.g. asymptomatic infections[39,42]. However, whilst this methodology is likely to be increasingly used in future, it is currently in its infancy and a large proportion of current viral knowledge is still the result of potentially biased top-down approaches.

The novelty of our approach lies in the separation of perspectives - by isolating the viral, mammalian and network perspectives we were able to further our understanding of mammalian hosts of known viruses in a number of ways. Firstly, our framework integrated local (mammalian and viral) and global (network) approaches. Our locally trained mammalian and viral models enabled the exploration of the effect, by means of variable importance, of a comprehensive set of mammalian and viral traits. We were able to measure the relative influence each of our

mammalian features had on each multi-host virus; and conversely, the influence each of our viral features had on each mammal (with two or more known viruses). This facilitated the aggregation of variable importance by, for instance, viral or mammalian taxonomy, which in turn illustrated differences in how these features influenced our models. For example, when aggregated at genus level, we found that body mass and a larger proportion of plants in the diet had higher influence on our models for Orbiviruses, which are known to infect ruminants and horses (7 species, median values = 90.97 and 86.83, respectively); whereas longevity, and weaning age were more influential to Ebolavirus models (5 species, 94.82 and 91.42, respectively). Uniquely, we incorporated geospatial features extrapolated from an extensive collection of global data on climate, environmental, agricultural, and mammalian diversity variables. The importance of these varied across our viral models. For instance, in coronaviruses, mean human population was more important for Beta-coronaviruses (83.38) than Alpha-coronaviruses (65.65). From the mammalian perspective, phylogenetic and ecological distances to known hosts were the most influential across all models. The importance of maximum phylogenetic breadth varied across families within the same taxonomic order. For instance, in rodents, it ranged from 89.08 (median) in Sciuridae (14 species) to 48.83 in Muridae (37 species). Local, species-level variable importance further enhances the utilisation of our approach to targeted surveillance, by enabling flexible aggregation of results from individual species to entire groups and orders.

Secondly, we consolidated these viral and mammalian traits with network topological features, expressed in terms of counts of potential motifs. We measured variable importance of our topological features and found that the likelihood of an association increased the more it featured in motifs linking its mammalian host with a virus with a wide host range (M4.1 and M5.1). Similarly, an association was more likely to be predicted by our network perspective models the more it featured in motifs linking a mammal with a wide range of viruses (M4.6 and M5.20), but the influence of these motif-features was not as high as the previous two. More complex motif-features (e.g. M4.4, M5.5, M5.9, M5.12, and M5.19) had a negative influence: the more an association featured in them, the less likely it was to be predicted by our models. This could be because these motifs indicate a separation between the known host range of the focal virus and

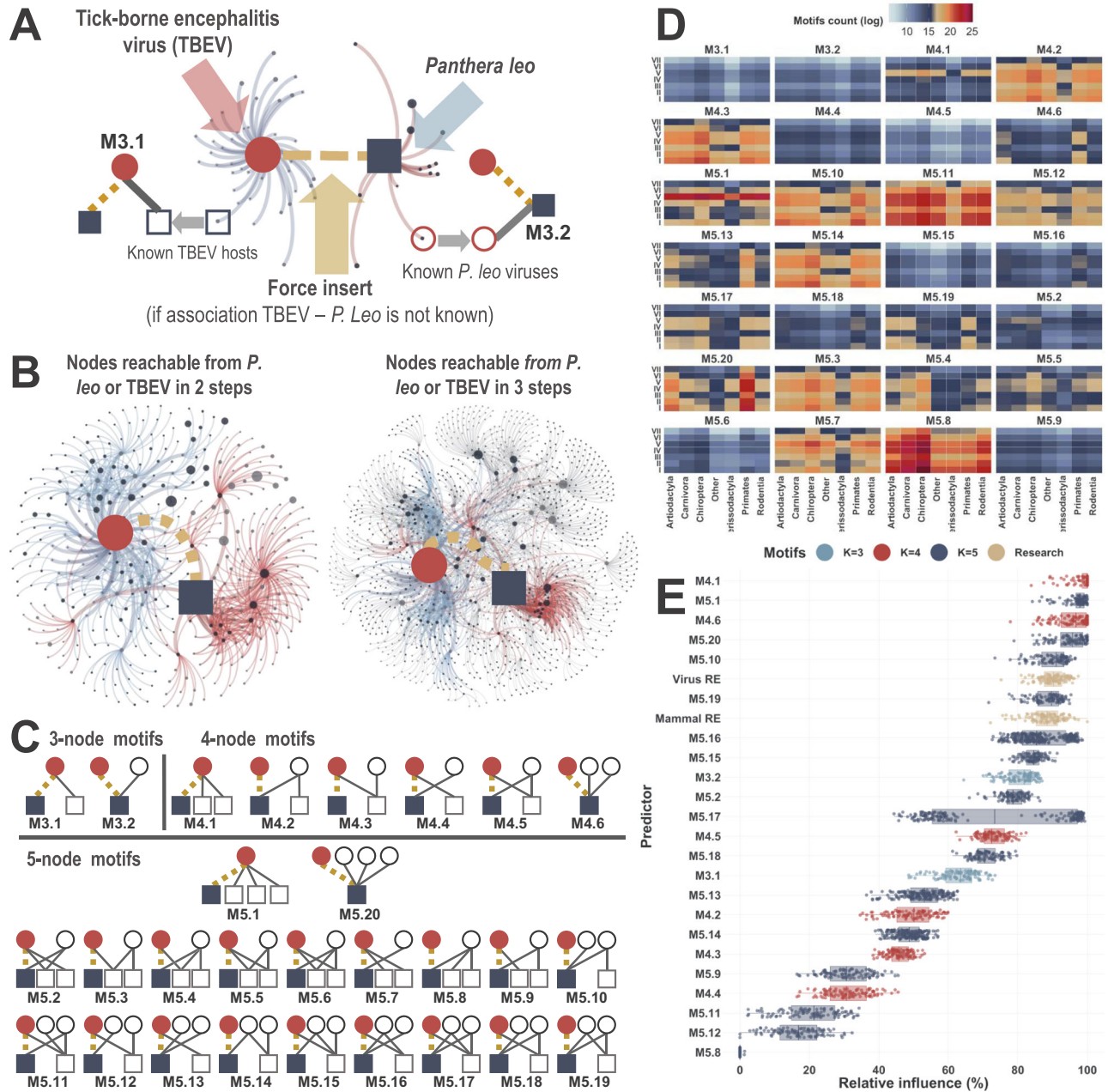

**Fig. 4 The network perspective - potential motifs (subgraphs) in our virus-host bipartite network. A** The concept of potential motif. The association TBEV-*P. leo* is a forced insertion into the network prior to calculating motifs for the association. **B** Motifs space: networks represent 2 steps and 3 steps ego networks (union) of host (here *P. leo*) and virus (TBEV). 1, 2 and 3 step ego networks comprise the counting space for TBEV-*P. leo* potential motifs. Dark grey nodes represent viruses, light grey nodes represent hosts. Size of nodes is adjusted to represent overall number of hosts or viruses with known associations to the node. Red edges represent nodes reachable from the mammal (*P. leo*) in 1 or 2 steps (links). Blue edges represent nodes reachable from the virus (TBEV) with 1 or 2 steps (links). Humans and rabies virus were excluded from these networks. **C** 3, 4 and 5-node potential motifs in our virus-host bipartite network. Circles represent viruses and squares represent mammals. Red circles represent the focal virus (v), and blue squares represent the focal mammal (m) of the association v-m for which the motifs are being counted (dashed yellow line). This association has two states: either already known (documented in EID2), or unknown. Grey lines illustrate existing associations in our network. **D** Motifs counts. Heatmap illustrating distribution of motif-features (counts of potential motifs per each focal association) in our bipartite network, grouped by mammalian order and Baltimore classification. The counts are logged to allow for better visualisation. **E** Variable importance (relative contribution) of motif-features (variables) to our network perspective models (SVM-RW). Motifs (subgraphs) are coloured by the number of nodes (*K* = 3, 4, 5). Boxplots indicate median (centre), the 25th and 75th percentiles (bounds of box) and inter quantile range (whiskers). Points represent variable importance in individual runs (*n* = 100). Research effort into both viruses and mammals is included as independent variables in our network models (coloured in yellow).

the focal mammal, and vice versa. For instance, higher counts of M5.19 suggest that, in general, there are no indirect pathways between the focal virus and mammal, despite the mammal featuring in several such pathways. Thus, higher counts of M5.19 might indirectly indicate that the focal virus is known to affect different types of hosts (e.g. different taxa).

Thirdly, our voting approach, despite being more conservative than its components (Supplementary Results 2, 4–6), was able to bridge a significant gap in our knowledge of mammalian hosts susceptible to included viruses (>18,000 associations between wild and semi-domesticated mammalian species and known viruses). Furthermore, our voting approach outperformed each of its constituent perspectives, and any combination of two perspectives, across all included metrics. The estimated improvements in performance metrics are essential, particularly for the application of our approach to targeted surveillance, because they indicate that in addition to its ability to detect documented associations very well, we have more confidence in predicted novel (unknown) associations (better F1-score) compared with results derived from any individual perspective, or by joining any two perspectives. Additionally, the results of our approach align with recent advances in the field of predicting novel hosts of known viruses, which all predict an increase in the host range[2,4,17,20,33,43,44]. For instance, we predict 44 novel associations between bats and Filoviruses (total of 60), which is a more conservative estimate than recent studies[43]. For flaviviruses, we predict 85 species of primates to be hosts to both zika and yellow fever viruses (20 species when voting with the 90th percentile across our 3 perspectives, we predict 20 primates to be hosts of both viruses) compared to 21 predicted in recent work[44]. Despite the large number of predicted, potentially novel, associations, the fact that our predictions can distilled to the level of individual virus or a mammalian species, makes our approach suitable for targeted surveillance per host or virus, or groups therein.

There remains, however, key areas for further improvement. We differentiate between two types of unknown virus-mammal associations: 1) associations between a known virus and a potentially susceptible mammalian host of this virus: known-unknowns; and 2) completely unknown viruses associated with a host but are not yet discovered: unknown-unknowns. Our approach aimed at the first type: we included as much information as available on known viruses and their susceptible mammalian species to predict associations between wild and semi-domesticated species and our viruses. In the case of species-rich mammalian orders containing sufficiently studied species (e.g. *Primates*, *Carnivora*), a higher proportion of their currently known viruses are likely to have been found. Hence, our approach was able to make predictions for wild and semi-domesticated (medium to under-studied) species in those orders. However, for mammalian orders with fewer species, and where those species are under-studied, there are more likely to be more unknown-unknowns, therefore a larger proportion of their viruses would not be predictable by our approach or other approaches.

Our approach also has limitations with regards to known-unknowns; we acknowledge that it does not entirely ameliorate the impact of research effort (Supplementary Figs. 10–14). Whilst our models did not necessarily over predict for heavily studied mammalian species, particularly humans and economically important domesticated animals, it predicted more known-unknowns for well-researched mammals (Supplementary Figs. 10–11, 14). The effect of research effort into viruses is more prominent, with our approach predicting significantly more potentially susceptible mammalian species for heavily studied viruses such as Influenza A virus and Rotavirus A (Supplementary Figs. 12–14). In other words, our approach cannot fully distinguish between two possible reasons for a mammal having

few virus associations: 1) the virus has never been observed in the mammal (due to research effort), and 2) the mammal is biologically not susceptible. One potential field-wide solution to this problem would be the inclusion of known-unsusceptible associations. This could potentially mitigate a large effect of 'research effort' related issues as well studies species would generally also have larger numbers of known-unsusceptible associations, which could tend to balance the effect. However, there are many reasons why this cannot be used at present, including: negative results are less likely to be published, especially for relatively under-studied and wild species; no resource of unsusceptible associations currently exists beyond review articles capturing a small number of either viral or mammalian species; and practical difficulties proving species-wide unsusceptibility to a given virus.

Prediction of unknown and novel viruses and their potential threat to humans, livestock and wildlife is an increasingly important and active research area. Where an established virus is increasing its geographical range (e.g. due to climatic or demographic factors), then our framework provides powerful means to assess potential hosts it has yet to come into contact with. The identification of these hosts is exceedingly important, as viruses continue to move across the globe via complex transmission cycles featuring migratory animals[45], legal and illegal trade in animals[46,47], unknown hosts (in various taxa, including non-mammalian hosts), bridge vectors[2], and reverse zoonoses[48]. However, for completely novel or never-studied viruses, our approach cannot predict potential associations due to lack of viral and network traits: an example is SARS-CoV-2; our pipeline could not have predicted its host association when it first emerged, but subsequent study of the virus, its traits and its observed hosts allows for prediction of its unobserved host associations[49]. Future work may be able to enhance the predictive power of our approach by incorporating more diverse viral traits, particularly in terms of detailed genetics[9] and in terms of geographical distribution and associated features of the virus as highlighted in previous work[50,51]. Integration of predictors of host-virus interactions such as the existence of particular viral receptors in host cells would also greatly benefit our models and create a fourth perspective that could be added into our framework.

Finally, a further separation of perspectives could also be achieved by incorporating arthropod vectors or intermediate hosts, or different classes of pathogens and hosts, particularly birds. Future integration of avian species into our network could potentially increase predictive power and explainability of our approach, particularly in relation to the ecology of viruses for which birds are known to be important reservoirs or amplifying hosts (e.g. flaviviruses such as West Nile and Japanese encephalitis, and influenza viruses). The incorporation of birds into our network component will enable quantification of yet-uncaptured important pathways in which birds play central roles. However, such integration will first require establishing a more complete picture of avian viruses and their hosts – the number of associations we were able to capture for avian species was 2,525 between 1,251 bird and 306 virus species (~40% of the total number of mammalian associations in this study). This could be achieved either by deeper mining of existing sources or by developing separate predictive pipelines focusing solely on birds.

In this study we attempted to expand our knowledge of viral host ranges by predicting the unknown hosts of known viruses. We applied a divide-and-conquer approach which separated viral, mammalian and network features into three unique perspectives, each predicting associations independently to enhance predictive power. We predicted over 20,000 unknown associations between known viruses and mammalian hosts, suggesting that current knowledge greatly underestimates the number of associations

between wild and semi-domesticated mammals. Completing the picture of virus-host interactions can help identify and mitigate current and future zoonotic and animal-disease risks, including spill-over from animals into humans.

## Methods

**Virus-host species associations**. Species-level virus-mammal associations were extracted from the ENHanCEd Infectious Diseases Database[10]– EID2 (version from December 2019). EID2 automatically mines information on pathogens (of any taxa), their hosts and locations from two sources: meta-data accompanying nucleotide sequences (hereafter sequences) published in Genbank[52,53]; and 2) titles and abstracts (hereafter TIABs) of publications indexed in the PubMed[54]. At time of extractions, EID2 has collated information from >7 million sequences (and processed 100 M + sequences), and >8 million TIABs. EID2 imports names of organisms (here viruses and mammals), and their taxonomy from the NCBI Taxonomy database[55]. It also extends these names with an exhaustive, expertly curated, collection of alternative and common names. These names are utilised to disambiguate hosts and pathogens in sequence meta-data and TIABs using inclusion and exclusion terms[10]. Evidence collated from TIABs is considered likely if it exceeds a given threshold (usually ≥ 4 publications). For the vast majority of stored organisms, EID2 follows the NCBI definitions of 'species' and 'subspecies', with unclassified and uncultured species being denoted as 'no rank'. For the purposes of this study, we recursively aggregated virus-mammal associations – a mammal that was found to host a strain or subspecies of virus was considered a host of the corresponding virus species (and vice versa). We further checked each of these species level associations for accuracy and to eliminate laboratory-produced results. This resulted in 6331 associations between 1896 viruses and 1436 terrestrial mammals. The support of these associations in EID2's evidence base was as follows: 22.79% had publication and sequence evidence; 33.03% were supported by nucleotide sequence only, and 44.18% were supported by evidence extracted from TIABs. The nature of this evidence was as follows: 70.48% of associations were strongly supported by sequence, isolation, or PCR evidence; 29.52% were supported by serology-only evidence. Of the total number of associations inferred from publication-only evidence, 66.82% were supported by serological evidence. We trained our pipelines with associations obtained from both sources; this is because serology is a standard means of determining previous viral infection in an individual. Isolation cannot detect an infection that has since been cleared by the host's immune system. Hence isolation and serology have different applications, and both should be utilised to get a more complete picture. Both sequencing and serological methodologies vary in their sensitivity and specificity depending on the virus clade. Both sequencing and serological methodologies vary in their sensitivity and specificity depending on the virus clade, with neither being superior in all cases[56–58]. Consequently, we chose to present the results using both isolation and serology in the manuscript. However, to account for possible variations in the strength of our evidence base, we trained a separate predictive pipeline including only those associations supported by sequence evidence (55.82% of total); Supplementary Results 8 summarise predictions of this pipeline; full results are included in our data release (see below).

**Multi-perspective framework to predict unknown virus-mammal associations**. We transformed our species-level virus-mammal associations into a bipartite network in which nodes represent either virus or mammal species, and links indicate associations between mammalian and viral species. Our bipartite virus-mammal network is sparsely connected – roughly 0.23% of potential associations are documented in EID2, despite it being the most comprehensive resource of its kind. This sparsity is more evident in wild and semi-domesticated species where only 0.182% of potential associations are observed. We treated the problem of bridging this gap in our knowledge of virus-mammal associations as a supervised classification problem of links in the bipartite network. In other words, we aimed to predict unknown associations between known viruses and their mammalian hosts based on our knowledge to date of these species. Each possible virus-mammal association is predicted three times as follows.

1 – From the mammalian perspective: For each mammal in our network, given a set of features (predictors) comprising viral traits (e.g. genome, transmission routes) – Table 1, what is the probability of an association forming between this mammal and each of the 1,896 virus species?

2 – From the viral perspective: For each virus species found in our network, given a set of features (predictors) encompassing mammalian phylogeny, ecology, and geographical distribution – Table 2, what is the probability of an association forming between this virus and each of 1,436 terrestrial mammals?

3 – Form the network perspective: Given a set of topological features representing the bipartite network expressing most of our knowledge to date of virus-mammal associations, what is the probability of an association forming between any virus and any mammal in our dataset ($n = 1,896 \times 1,436 = 2,722,656$ possible associations)?

Our framework trained and selected a set of supervised classifiers in each of the above perspectives as discussed below. It then consolidated the results of the best performing classifiers using voting whereby an unknown (potential or unobserved/undocumented) association was selected if it was predicted by at least two of the

three perspectives. This is because each of our perspectives focuses on a particular aspect of virus-host associations. From the mammalian perspective, and for every included mammal, the probability of a virus affecting/associating with this focal mammal is quantified based on our knowledge of the viruses found in this mammal to date. Similarly, from the viral perspective, the probability of the virus infecting/associating with included mammalian species is quantified based on our knowledge to date of known hosts of this virus. The final perspective enables generation of predictions based on the topology of the network linking all included mammals with all included viruses. Thus, our three perspectives capture all aspects of viral-mammalian association without biasing toward one aspect.

Our framework is flexible, in terms of machine-learning algorithms selected, classifiers trained, and features engineered for each perspective. It avoids overfitting as it approaches the problem from various perspectives, and effectively consolidates ensembles of classifiers trained on subsets of the underlying data. In addition, no constituent model of our framework has been trained with all available data at any time. Finally, our framework enables the incorporation of hosts where only one virus has been detected to date (via perspectives 2 and 3), and viruses where only one host has been discovered (via perspectives 1 and 3).

**The local approach – the mammalian and viral perspectives**. Our mammalian and viral perspectives generate "local" predictions for hosts and viruses, respectively. These local predictions are derived by training a suite of models for each host (with two or more known viruses), and virus species (with two or more known mammalian hosts), as described in subsequent sections. In other words, each mammalian species has its own "local" suite of models, trained using viral traits (Table 1), to predict viruses which could associate with this host. Similarly, each selected virus has its own set of models, trained using mammalian features (Table 2), to predict mammalian hosts which are potentially susceptible to this virus. The reason for predicting locally (per host, or virus) is two-fold: 1) Variations of host susceptibility, viral host range: traits (features) determining, for instance, mammalian species susceptibility to West Nile virus, are potentially different to those affecting these species' susceptibility to Bovine immunodeficiency virus. Hence, by training these models locally, we are able to ascertain the influence of these traits on each host, and each virus. 2) Class balancing: we synthesised new positive training instances for each of our hosts, based on the traits of their known viruses Likewise, we synthesised new positive instances for each of our viruses, based on the traits of their known mammalian hosts (as discussed below).

**The network perspective - topologically derived network features of virus-mammal associations**. In contrast with our mammalian and viral perspectives, the network linking known viruses with their mammalian hosts presents a "global" view of how these viruses are shared amongst their mammalian hosts. Here we capture the topology of this bipartite network by means of counts of potential motifs[21] (Fig. 4A, C). These motifs capture important indirect pathways between viruses and their mammalian hosts. These pathways vary from simple generalisations capturing whether a virus has wide range of hosts or not (M3.1, M4.1, and M5.1), or if the mammal is exposed to many viruses (M3.2, M4.6, M5.20), to more complex pathways (e.g. two host species sharing 80% of their viruses with each other; three viruses sharing 50% of their hosts with each other). These pathways might indicate if an unknown association is more likely to exist in nature or not, and are only capturable, and most importantly quantifiable (Fig. 4D), at the global level as encapsulated by our network perspective.

Transforming these pathways into features from which supervised machine-learning algorithms could learn, enables us to make predictions directly from the network structure. Here, counting of potential motifs is limited to the 3-step ego network of both virus and host – the network comprising nodes which can be reached in 3 steps (links) or fewer from each focal node (nodes comprising the focal association (Fig. 4B).

We generated a features-set comprising the counts of potential motifs for all associations (Fig. 4D) and trained several machine-learning algorithms with this dataset (plus research effort) as detailed in following subsections. Motifs are usually associated with specific frequency thresholds[23]. However, here we follow previous work[21] in removing this restriction. We simply counted the number of occurrences of potential motifs of each focal association, and then let the machine-learning algorithms detect which motifs were particularly important to the problem of predicting links in our network (Fig. 4E).

**Research effort**. We incorporated research effort on mammal and virus species into our network perspective models. This is because it is through this perspective that predictions are made for all hosts and all viruses at the same time, and where the effect of research effort into both the hosts and viruses can be measured and corrected for adequately and simultaneously. We calculated research effort as the total number of sequences and publications of each species as indexed by EID2[10]. In addition, we trained a separate pipeline in which the research effort into our hosts was included as predictive feature in each constituent viral perspective model; and the research effort into our viruses was included in each constituent mammalian perspective model. Agreement between training constituent models with and without research efforts in mammalian and viral perspective was 99.7% [99.9%–99.2%] (values in bracket are confidence intervals derived from predictions

CI). Cohen's Kappa = 0.86 [0.85–0.89] (Kappa range: 0-1). Results of this pipeline are listed in Supplementary Results 1. Detailed validation of both pipelines is listed in Supplementary Results 2.

**Multi-perspective prediction of virus-mammal associations**. As highlighted above, our framework comprised three perspectives: mammalian, viral and network. Each of these perspectives trained a set models with different features (Tables 1 and 2, and Fig. 4 respectively), and hence required its own pipeline as described below (Supplementary Note 5).

**Mammalian and viral perspectives**

*Class balancing.* On average each virus in our dataset affected 3.45 mammals (~0.240% of the 1436 mammals in our models), and each mammalian host was affected by 4.41 viruses (~0.241% of the 1833 viruses in our models). This presented an imbalance in our data, whereby a small percentage of instances are actualised. We dealt with this issue in two ways: first we excluded any virus ($n = 1281$) which was found in only one mammal species from our virus models pipeline (viral perspective), and we excluded any mammal ($n = 758$) which is only affected by one virus from our mammal models pipeline (mammalian perspective). Second, we deployed SMOTE - Synthetic Minority Over-sampling Technique[59,60] to rebalance the classes prior to training each of our viral ($n = 8 \times 556$) and mammalian ($n = 8 \times 699$) models. SMOTE synthesises new minority class instances from existing minority instances using a variation of k-nearest neighbour algorithm. The SMOTE algorithm then over-samples from the minority instances (original and synthesised) and under-samples from the majority class to create a balanced training set. All class balancing was achieved using caret R package[63] (R version 3.6.2).

*Classification algorithms.* For each mammal and each virus selected above we trained 8 classification algorithms (Supplementary Table 4): Model Averaged Neural Network (avNNet), Stochastic Gradient Boosting (GBM), Random Forest, eXtreme Gradient Boosting (XGBoost), Support Vector Machines with radial basis kernel and class weights (SVM-RW), Linear SVM with Class Weights (SVM-LW), SVM with Polynomial Kernel (SVM-P), and Naive Bayes. These classifiers offer a varied subset of plethora of classifiers available for experimentation (over 179 classifiers categorised into at least 17 families[61]), and were selected due to their robustness, scalability[61], and their potentially good performance with imbalance data classification[62]. All models were trained and optimised using caret R package[63] (R version 3.6.2) as described below.

*Training and tuning.* each of the above models was trained with 10-fold cross validation (10 repeats). This validation method works by splitting training data into 10 random samples, each sample is held out in turn, and the model is trained on the remainder groups. The model's prediction for the existence or absence of the mammal-virus associations in the held-out group are used to construct confusion matrices and calculate an optimisation metric (here Area Under the ROC Curve, AUC for short). The optimisation metric is used to select best model in the validation process.

We adopted an adaptive resample approach[64] to tune the hyper-parameters of our models. This approach resamples the tuning parameter grid in a way that concentrates on values that are the in the neighbourhood of the optimal settings (adaptive). Due to the large number of classifiers trained in our framework this adaptive approach allowed us to find optimal (or near optimal) values of the hyper-parameters of each included machine-learning algorithm without relying on the nominal resampling process whereby all the tuning parameter combinations are computed for all the resamples before a choice is made about which parameters are good and which are poor.

*Classifier selection strategy.* We computed three performance metrics based on the median predicted probability across each set of replicate models: AUC, true skills statistics (TSS) and F1-score (Supplementary Table 5). The best performing classifier per each virus or mammal, across all measures, was included in our multi-perspective final model (Supplementary Results 4 and 5).

*Confidence intervals.* In order to allow us to incorporate uncertainty arising from variations in SMOTE resampling technique and resulting training sets, and to generate empirical confidence intervals (90%), hyper-parameters of best performing models were carried across to train 50 replicate models for each best performing mammalian or viral model. In other words, we generated a bragging (i.e. median) ensemble for each selected host or virus, and the resulting prediction was carried to our multi-perspective final model.

**Network perspective**

*Class balancing.* Our bipartite virus-mammal network is sparsely connected with 6331 documented associations out of 2,722,656 possible associations (0.23%). Due to this we implemented strict under-sampling: whereby balanced samples drawn at random (without replacement) from the set of all potential virus-mammal

associations. Each sample comprised 2000 instances (1000 positive (known) and 1000 unknown virus-mammal associations.

*Training & tuning.* We trained the same selection of algorithms as above with balanced sets (2000 instances each) using 10-fold cross validation with adaptive resampling to optimise AUC. We repeated this process 100 times to generate a bragging ensemble of predictions (derived as probabilities) of these replicate models. We calculated empirical confidence intervals (90%) of the ensemble probabilities across the 100 replicate models.

*Classifier selection strategy.* We selected the bragging ensemble which obtained the best overall performance metrics (AUC, F1-Score and TSS) when applied to all available associations. The predictions of the best overall ensemble were incorporated into our final model (SVM-RW - Supplementary Results 6).

**Performance assessment**. We trained the constituent models of each perspectives with a stratified random training set comprising 85% of all data ($n = 2,315,391$ with 5377 known virus-mammal associations). The processes described above were repeated with training set only, and performance was measured against the held-out test set (15% of all data, $n = 407,265$ with 954 known virus-mammal associations). Performance metrics obtained through this assessment are reported above and in Supplementary Results 2. Additionally, we performed a complementary test to assess the ability of our model to predict systematically removed virus-mammal associations (Supplementary Results 3).

**Variable importance**. we calculated relative importance (influence or contribution) of viral (Table 1), mammalian (Table 2), and network features (Fig. 4C) to each model in our three perspectives. Due to the selection strategy implemented in our viral and mammalian perspectives, whereby models from 8 different algorithms were selected, we computed the importance of these features using a model-independent filter approach via a ROC curve analysis conducted on each predictor (as implemented in the caret package[63]).

**Reporting summary**. Further information on research design is available in the Nature Research Reporting Summary linked to this article.

## Data availability

Virus-mammal species-level associations were obtained from the ENHanCEd Infectious Diseases Database (EID2). Viral, mammalian and geospatial data were obtained from open-access data sources. These sources are listed in detail in Supplementary Notes 1–3 of the Supplementary Information file, and their DOIs are provided in the Supplementary References. Data used can be found here: https://doi.org/10.6084/m9.figshare.13270304, with the exception of mammalian presence shapefiles and raw climate data (due to their large size) - these data can be obtained from the authors or directly from the sources listed in the Supplementary Information file. Final and intermediate (perspective) predictions of our approach, and predictions obtained using only sequence-evidence are also made available (https://doi.org/10.6084/m9.figshare.13270304).

## Code availability

All codes used in our analyses are made available via figshare (https://doi.org/10.6084/m9.figshare.13270304).

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

## Acknowledgements

MW acknowledges support from BBSRC and MRC for the National Productivity Investment Fund (NPIF) fellowship (MR/R024898/1). Establishment of the EID2 database was funded by a UK Research Council Grant (NE/G002827/1) to MB, as part of an ERANET Environmental Health award to MB; subsequently, it has been further developed and maintained by BBSRC Tools and Resources Development Fund awards (BB/K003798/1; BB/N02320X/1) to MB, and the National Institute for Health Research Health Protection Research Unit (NIHR HPRU) in Emerging and Zoonotic Infections at the University of Liverpool in partnership with Public Health England and Liverpool School of Tropical Medicine.

## Author contributions

M.W. compiled the data, designed, and produced the analyses. All authors contributed to the study design. M.W. and M.B. established the EID database. M.S.C.B. and K.J.S. provided viral and network expertise. All authors contributed equally to the writing of the manuscript.

## Competing interests

The authors declare no competing interests.
