## [Peer Review File · Nature Communications]

Divide-and-conquer: machine-learning integrates mammalian and viral traits with network features to predict virus-mammal associations – Supplementary Information File

Maya Wardeh*, Marcus SC Blagrove, Kieran J. Sharkey, Matthew Baylis
 *maya.wardeh@liverpool.ac.uk

Supplementary Note 1 – Virus-mammal interactions data

Supplementary Table 1 – Baltimore classification of mammalian viruses included in this study. Virus classification followed NCBI taxonomy¹ (Dec/2019).

	Group	Virus species	Host species (EID2) ²	Description
DNA	Group I	726	491	dsDNA viruses - double-stranded DNA viruses (e.g. herpesviruses)
	Group II	273	160	ssDNA viruses - single-stranded DNA viruses (e.g. circoviruses)
RNA	Group III	46	156	dsRNA viruses - double-stranded RNA viruses (e.g. rotaviruses)
	Group IV	427	511	(+)ssRNA viruses - positive-sense single-stranded RNA viruses (e.g. Zika virus, Yellow fever virus)
	Group V	360	1051	(-)ssRNA viruses - negative-sense single-stranded RNA viruses (e.g. Ebola virus, Influenza A virus)
Retro-transcribing	Group VI	57	182	ssRNA-RT viruses - (+ strand or sense) RNA with DNA intermediate in life cycle (e.g. HIV1)
	Group VII	7	41	dsDNA-RT viruses - DNA with RNA intermediate in life cycle (e.g. Hepatitis B virus)

Domestication level of mammals: We classified the domestication status of our mammalian hosts into four levels: wild mammals ($n=1326$), semi-domesticated mammals ($n=81$), domesticated mammals ($n=28$), and humans ($n=1$). Supplementary Table 2 lists our domesticated mammals. Our semi-domesticated mammals group included $n=29$ of each *Carnivora* and *Ruminantia*, 13 *Rodentia*, 3 *Perissodactyla*, 2 of each *Proboscidea* and *Tylopoda*, and one of each *Diprotodontia*, *Eulipotyphla* and *Suina*.

Supplementary Table 2 – Domesticated mammals included in our analyses.

	Species	Species
Artiodactyla	16	Bison bonasus , Bos frontalis , Bos grunniens , Bos indicus , Bos javanicus , Bos taurus , Bos mutus , Bubalus bubalis , Bubalus carabanensis , Capra hircus , Ovis aries , Camelus bactrianus , Camelus dromedaries , Lama glama , Lama pacos , and Sus scrofa
Carnivora	3	Canis lupus familiaris , Felis catus , and Vulpes vulpes .
Perissodactyla	3	Equus asinus , Equus caballus , and Equus asinus x caballus
Rodentia	5	Cavia porcellus , Mesocricetus auratus , Mus musculus , Rattus norvegicus , and Rattus rattus .
Lagomorpha	1	Oryctolagus cuniculus

Supplementary Note 2 – Viral Feature Space (Viral traits & features)

Virus genome and capsid: We classified the genome of each virus as RNA (binary factor, no=DNA); retro-transcribing (binary factor, NCBI taxonomy¹ - Dec/2019); negative-sense (binary factor, yes = negative-sense; and positive-sense (binary factor, yes = positive -sense, NCBI taxonomy¹). RNA viruses adapt faster³, and are generally more fragile (cannot survive as long outside of the cell). Retroviruses are generally very conserved⁴, and have to enter the nucleus⁵ and insert into the genome, these additional steps may require specificity and limit range. Sense affects replication cycle and range of host enzymes needed.

With regards to the genome architecture and organisation we checked if the virus has circular or linear genome (binary factor, no=linear, obtained from ViralZone⁶- Dec/2019), as this attribute affects replication and translation. Rolling circle replication and translation are common with circular genomes, negating the need to re-enlist host enzymes⁷, therefore possibly affecting host range as less specificity may be required. In addition, we noted if the virus was monopartite (has a single nucleic acid molecule protected in a shell made of proteins) or segmented (divided into two or more nucleic acid segment) (binary factor, no=monopartite, obtained from ViralZone⁶). Segmented viruses can undergo recombination if two strains of the same virus infect a cell (e.g. influenza hemagglutinin & neuraminidase recombination⁸). This in turn can lead to host range changes of segments of the genome. In practice, there exists a third class of viral architecture - multipartite viruses. These viruses have their genome divided into two or more nucleic acid segment (similarly to segment viruses), but these segments are each packaged into separate virus particles. We ignored multipartite viruses in this study due to them being very rare and poorly understood⁹.

Regarding capsids, we indicated if the virus is enveloped or not (binary factor, obtained from ViralZone⁶). Envelopes are usually derived from the host cell membrane; this can help them avoid host immune system. They may limit range by providing antigens for other immune systems. The envelopes are very sensitive to the external environment, and enveloped viruses often require to be directly transferred between hosts; finally, because the envelope is made from the current host's cell membrane, it will change upon infection of a new host, making the virus rapidly adaptable¹⁰.

GC-content (guanine-cytosine content, obtained from ViralZone⁶ and NCBI Genbank^{11,12}- Dec/2019), the percentage of a nucleotide sequence that is made up of either guanine or cytosine bases of each virus, and the average genome size (in bases) were also obtained. GC hydrogen bonding is stronger than AT/U, high GC content usually leads to higher thermo-stability of the genome¹³, including single stranded genomes which often self-anneal. This may affect longevity outside of hosts, and replication inside of ectothermic hosts. Genome size is indicative of many aspects of the virus such as complexity, DNA/RNA, and replication type. Virus species not captured directly were assigned genus level traits.

Virus replication, release, and cell entry: We collated information on the replication site of the virus. We expressed these data as a binary factor indicating if the virus replicates in the cytoplasm (versus nucleus replication). Replication site is linked to RNA/DNA genome – if a virus has a DNA stage it must replicate in the nucleus. This creates extra barriers to overcome for entry to the nucleus and may restrict host range.

We classified the release of the virus into three broad categories: budding, lysis, or other. The mechanism of release affects aspects the rate of virus production, cell life-span and means of presentation to the immune system¹⁴, each of these aspects could influence the virus host range. Finally, we recognised that availability of receptors influences potential host range, therefore we broadly categorised the mechanism of virus cell entry into 4 categories: cell-receptor endocytosis, clathrin-mediated endocytosis, glycoprotein-mediated or other. Replication, releases and cell entry

information were obtained from ViralZone⁶ and ICTV¹⁵ (Dec/2019). Virus species not captured directly were assigned genus level traits.

Transmission routes: We categorised major transmission routes of 1,897 viruses in 6,445 associations with 1,471 mammals (including aquatic species) as follows.

direct transmission – by direct contact, via skin, broken skin or droplets (unique virus species $n_v=1,168$, mammalian hosts $n_m=1305$, associations= $4,389$); sexual transmission ($n_v=273$, $n_m=405$, associations= $1,154$); vertical transmission – mother to child, or via breast-milk ($n_v=252$, $n_m=445$, associations= $1,161$); indirect transmission – e.g. via secretions/excretions/tissues ($n_v=602$, $n_m=1260$, associations= $3,224$); ingestion – including water ingestion and faecal-oral routes ($n_v=848$, $n_m=1,013$, associations= $2,792$); inhalation – via droplet or airborne particles or dust ($n_v=641$, $n_m=612$, associations= $2,029$); environmental – through fomite, contact with environment ($n_v=372$, $n_m=633$, associations= $1,596$); and vector – strictly via arthropod vector such as ticks or mosquitoes ($n_v=194$, $n_m=377$, associations= $1,019$).

We adopted a three-fold strategy to capture major transmission routes of our viruses as follows:

1. Title and abstract (TIABs) mining: we utilised EID2 to extract TIABs of PubMed papers linked to single virus species (i.e. excluding TIABs with multiple viruses). These TIABs were subsequently classified via keyword matching into the transmission routes described above. The TIABs were further checked manually to ensure correctness and to remove transmission routes outside mammalian species (e.g. we removed instances of vertical transmission within arthropod vectors).
2. Manual extractions from online-sources: we manually extracted transmission routes of viruses for which no papers were identified by the previous step, as well as for routes not detected in the TIABs from various sources^{6,15,16}.
3. Within genus generalisations: finally, we assigned transmission routes to viruses not captured by the previous two steps, by taking a minimal agreement set of within-genus transmission routes.

For the purposes of this study, we utilised a simple multi-label classification of our viruses whereby routes were assigned either 1 or 0 value corresponding to whether the virus was found to be transmitted via the specified route as described above.

Supplementary Note 3 – Mammalian Feature Space (mammalian traits & features)

Phylogeny: Host phylogenetic distance has been found to drive sharing of pathogens, particularly viruses¹⁷⁻¹⁹, between mammals. We utilised a recent mammalian supertree²⁰ to calculate pairwise phylogenetic distance between each mammal-mammal pair. We then aggregated these values (mean) between each mammal species and the known hosts of each of our viruses. This measure indicated, per virus, whether a potential host species was phylogenetically close to or distant from the viruses preferred host range. Mammalian species for which could not be matched to this phylogeny was dropped from our analyses.

In addition, we calculated the evolutionary distinctiveness for each mammal using fair proportion²¹, as implemented in the R package *picante*²². Evolutionary distinctiveness quantifies how isolated a species is on its phylogenetic tree²³, and has been shown to correlate negatively with pathogen species richness²⁴.

Host traits: We compiled data on morphological and life-history traits, diet and habitat for our mammal species from online databases and literature^{17,25-29}. We selected the following traits for their known correlation with host-pathogen associations, and their wide availability. Body mass (g) represented morphological traits, as it proxies key features of metabolism and adaptation to environment. For life-history traits we included: maximum age (months), activity cycle, and migration²⁶. We included gestation period length (days), litters per year, litter size, weaning age (days), and age at sexual maturity (days), to represent the reproductive characteristics of our mammalian species. Reproductive traits could be viewed as proxies to within-host virus-dynamics and therefore may influence the viruses harboured by the host. Mammalian species not captured directly were assigned traits aggregated at genus level.

We included geographical area range²⁸ (in km²) as species with wider areas might be exposed to more viruses. We incorporated habitat utilisation²⁸ as multiple binary indicators of whether a species uses one or more of 14 natural and artificial habitats. We hypothesised that mammals utilising similar habitats might encounter similar viruses, and this in turn would increase the chances of being infected with these viruses.

We used the proportional use of 10 diet categories²⁷ to indicate the dietary preferences of mammals. We incorporated these categories as independent variables in our models as we assumed that similar dietary habit might associate with similar viral assemblage.

We included the above listed traits as independent variables in our virus perspective models. In addition, we utilised them to quantify the pair-wise ecological distance between each pair of mammals in our study. We based this distance on a generalised form of Gower's distance matrices^{30,31}. We incorporated these distances to compute the mean distance between each mammal and known hosts of each virus (and vice versa). Similarly, to the mean phylogenetic distance listed above, the mean ecological distance indicated, for each virus, whether a potential host species was ecologically close to or distant from the viruses preferred host range.

Mammalian geospatial features: The geographical distribution of host species influences the pathogens with which it might come into contact. Geography also correlates with other factors such as climate, natural environment, and agricultural practices. Climate has been shown to influence a number of human and domestic mammal pathogens (including viruses)³²⁻³⁴. Furthermore, climate indirectly affects certain groups such as vector-borne viruses (e.g. bluetongue and Zika) through the direct effect it has on the associated arthropod vectors^{35,36} and their competence. In addition to climate, other geographical factors such as land cover type, biodiversity (species richness), urbanisation and human population, and farming and agriculture practices have been found to influence certain categories of host-pathogen associations^{37,38}.

We obtained and processed species-presence maps of our mammals^{28,39,40}. We supplemented these with grids expressing climate⁴¹, mammalian diversity⁴², human population⁴⁰, land cover (including urbanisation)⁴³, agriculture^{43,44}, and distribution of livestock³⁹. This allowed us to generate the following geographical features of mammalian species:

1. We expressed climate in two features: mean temperature, and mean precipitation.
2. We quantified the diversity of natural land cover type (not directly associated with humans).
3. We quantified agricultural (including land-cover associated with humans e.g. managed vegetation) and farming practices (expressed in number of domesticated livestock and poultry in the species presence area) as entropies of the associated values in the species range.
4. We computed urbanisation as percent of urban land in the species presence area.
5. We summed human population in the species presence area.
6. Finally, we computed average mammalian diversity in the species presence area.

However, a limitation of our approach is that only presence/absence information were available for our wild species, rather than detailed density maps. This meant that our geospatial features were derived by equal weighting of the underlying features (e.g. land-cover), within the host range (presence), rather than by weighing by host density.

We obtained species-presence maps for majority of our mammalian species from IUCN²⁸. We extrapolated livestock (including horses) species-presence maps from recent global distribution maps^{39,45}. Finally, we inferred presence-maps for three domesticated species - dogs (*Canis lupus familiaris*), cats (*Felis catus*) and guinea pigs (*Cavia porcellus*) from gridded population of the world maps⁴⁰, by assuming they co-exist with humans where there is sufficient human populations ($n>100$). We used the same gridded population maps to extrapolate human species-presence map ($n>0$). All our geographical maps manipulation was done in QGIS.

We dropped any mammalian species for which no presence maps could be derived from our models trained in viral perspective.

We utilised several sources to extract geo-attributes of our mammalian species. Supplementary Table 3 lists these sources, features derived, and reason for inclusion in our analyses. Supplementary Figure 1 illustrates this process. We derived the following geo-features for our mammalian species:

1. Climate attributes:
 - a. Mean temperature: mean of monthly temperatures recorded in the species-presence area, averaged between years: 1900-2010⁴¹.
 - b. Mean precipitation: Sum of monthly precipitation recorded in the species-presence area, averaged between years: 1900-2010⁴¹.
2. Natural land cover diversity: calculated as Shannon's entropy of mean percent of land covered with each of the attributes listed in Supplementary Table 3 under category (Natural) Land-cover.
3. Agriculture and farming diversity: calculated as Shannon's entropy of mean percent of land utilised for the following categories: managed/cultivated vegetation, regularly flooded vegetation, cropland and pasture, and the sum of each of the livestock, horses and poultry listed in Supplementary Table 3.
4. Human population: human population in the species-presence area calculated by intersecting with a recent gridded human population dataset⁴⁰.
5. Urbanisation: mean percent of urban land in the species-presence area.
6. Mammalian diversity: mean mammalian diversity⁴² in the species-presence area.

Supplementary Table 3 - List of geographical predictor layers integrated within our framework.

	Geographical layer	Source	Resolution	Reason
(Natural) Land-cover	Evergreen/deciduous needle-leaf trees (%)	EarthEnv ⁴³	0°0'30"	Type of land cover has been associated with distribution of various mammals ⁴⁶ . It potentially increases chances of contact between mammalian reservoirs of different viruses.
	Evergreen broad-leaf trees (%)			
	Deciduous broad-leaf trees (%)			
	Mixed/other trees (%)			
	Shrubs (%)			
	Herbaceous vegetation (%)			
	Barren land (%)			
Agriculture & farming	Managed/Cultivated Vegetation (%)	EarthEnv ⁴³	0°0'30"	Livestock farming has been linked to emergence and cross-species transmission of number of viruses (e.g. Nipah virus, influenza viruses) ⁴⁷ .
	Regularly flooded vegetation (%)			
	Cropland (%)			
	Pasture (%)	HYDE ⁴⁴	0°5'	
	Cattle and buffalo (head count)			
	Sheep (head count)			
	Pigs (head count)			
Poultry (chicken and duck - head count)	GLW (Gridded Livestock of the World) ⁴⁵			
Human population		SEDAC ⁴⁰	0°5'	Urbanisation and human population density have been shown to be drivers of disease emergence and spill-over through wildlife-domestic-human interface ^{38,48,49} .
Urban land (%)		EarthEnv ⁴³	0°0'30"	
Climate		Mean temperature	CRUTS3 ⁴¹	0°5'
	Mean precipitation			
Mammalian diversity	Number of different mammalian species in a grid cell.	SEDAC ⁴²	0°5'	Mammalian species present in mammal rich areas might be exposed to diverse viruses ⁵⁰⁻⁵³ .

Supplementary Figure 1 - Geographical feature extraction. Mammalian species-presence maps were first extracted from our sources^{28,39,40} – step 1. These maps were then intersected with our geo-layers (Supplementary Table 3) – step2. This enabled us to derive values of our geo-attributes for majority of our mammalian species – step3, which we then summarised into the geo-features included in our models – step 4. Source of mammalian silhouette graphics is PhyloPic.org. Mock species-presence maps were hand-drawn by the authors.

Supplementary Note 4 – Potential motifs in bipartite networks

Supplementary Figure 2 – Motifs as features of bipartite networks (domestication status). Heatmaps illustrating distribution of motif-features (counts of potential motifs per each possible edge) in our bipartite network, grouped by domestication status of mammalian hosts and Baltimore classification of the viruses. Counts are logged to allow for better visualisation.

Supplementary Figure 3 – Motifs as features of bipartite networks (transmission routes). Heatmaps illustrating distribution of motif-features (counts of potential motifs per each possible edge) in our bipartite network, grouped by order of mammalian hosts and transmission route of the viruses. Counts are logged to allow for better visualisation.

Supplementary Note 5 – Multi-perspective framework to predict unknown virus-mammal associations

Supplementary Table 4 - List of machine learning algorithms (supervised classifiers) used in our models.

	Caret method	Base family	Summary
Model Averaged Neural Network (avNNet)	avNNet	Neural networks	The same neural network model is fit using different random seeds. All resulting models are then used for prediction, by averaging output probabilities from constituent models.
Stochastic Gradient Boosting (GBM)	gbm	Decision trees	GBM ⁵³⁻⁵⁶ fits a series of trees (weak classifiers) to random partition of the data, and aggregates the results sequentially (boosting).
Random Forest (RF)	ranger		RF algorithm ⁵⁷ constructs an ensemble (forest) of decision trees grown randomly from the input data. The randomness is twofold: 1) each tree is built from a random sample of the data. 2) at each node, a subset of features is randomly selected to generate the best split.
eXtreme Gradient Boosting (XGBoost)	xgbTree		Similarly to GBM, XGBoost ⁵⁸ constructs an ensemble from weak learners, typically decision trees. XGBoost has the additional advantages of speed; ease of use and parallelisation; and in some cases, higher predictive accuracy.
Support Vector Machines with Radial basis kernel and Class Weights (SVM-RW)	svmRadialWeights	Support vector machines	Support vector machines (SVMs) aim to find a hyperplane that best separates the features (predictors) into different domains (classes). Radial SVMs utilises Radial Basis Function Kernel (RBF), whereas polynomial SVMs adopts polynomial kernels. Linear SVMs uses linear kernels. SVMs with class weights have an additional built-in robustness to class-imbalance as they apply penalty to misclassification, with weights inversely proportional to class frequency.
Linear Support Vector Machines with Class Weights (SVM-LW)	svmLinearWeights		
Support Vector Machines with Polynomial Kernel (SVM-P)	svmPoly		
Naive Bayes (NB)	naive_bayes	Bayesian	NB classifiers assume that input features (viral, mammalian and network features in our case) are independent – they contribute independently to probability to the outcome class (a virus-mammal association in our case), regardless of any correlation between the features.

Supplementary Table 5 - Measures utilised to assess the performance of our 10-fold cross-validated classifiers. Bolded measures were used in classifier selection.

Confusion matrix		
		Detected
Predicted	1	0
1	A	B
0	C	D
Measure	Formula	Meaning
Sensitivity (recall)	$\frac{A}{A + C}$	Sensitivity is the percentage of actual positives (observed associations) that were correctly predicted. It indicates the percentage of 1s that was covered by the model.
Specificity	$\frac{D}{B + D}$	Specificity is the percentage of negatives (here unknown associations, not necessarily true negative) that were correctly predicted
Precision	$\frac{A}{A + B}$	Percentage of accurate predictions of the model
AUC	Area Under the ROC Curve	AUC is a threshold-independent measure of model predictive performance that is commonly used as a validation metric for host-pathogen predictive models ^{59,60} .
F1-score	$2 \times \frac{\text{Precision} \times \text{Recall}}{\text{Precision} + \text{Recall}}$	Captures the harmonic mean of the precision and recall. it is often used with uneven class distribution. Our approach is relaxed with respect to false positives (unknown associations), hence the low F1-score recorded overall. However, in our selection process, where two classifiers produced similar AUC and TSS statistics, the best performing on F1-score was selected (conservative approach).
TSS	Sensitivity + Specificity – 1	Use of AUC has been criticised for its insensitivity to absolute predicted probability and its inclusion of a priori untenable prediction ^{53,61} , we also calculated the True Skill Statistic (TSS) ⁶² .

Supplementary Figure 4 – Training and validation of mammalian perspective models. Training and test sets (85%, 15%) are split into ($n=699$) training and test sets, where each training/test pair contains associations of one mammalian species (e.g. humans) with all viruses in our input set ($n=1,896$). For each mammalian species, the training set is first balanced using SMOTE, then 8 classifiers are trained with this balanced set. These classifiers are then tested against the (unbalanced) test set, and the best performing (per each species) is carried forward to our multi-perspective framework.

Supplementary Figure 5 – Training and validation of viral perspective models. Training and test sets (85%, 15%) are split into ($n=556$) training and test sets, where each training/test pair contains associations of one mammalian species (e.g. humans) with all viruses in our input set ($n=1,436$). For each mammalian species, the training set is first balanced using SMOTE, then 8 classifiers are trained with this balanced set. These classifiers are then tested against the (unbalanced) test set, and the best performing (per each species) is carried forward to our multi-perspective framework.

Supplementary Figure 6 – Training and validation of network perspective models. 100 balanced samples (under-sampling) are drawn randomly from the training set (85% of all potential associations). Each sample (2,000 instances: 1,000 negative and 1,000 positive associations) is used to train 8 different classifiers (one of each of included algorithms - Supplementary Table 4). A Bragging (median prediction, probability) ensemble is then created for each algorithm. These ensembles are tested against the held-out test set (15%) and the best ensemble is carried forward to our multi-perspective framework.

Supplementary Results 1 – Incorporating research effort into models trained in the viral and mammalian perspectives

We separately trained constituent models of our mammalian and viral perspectives with research effort of viruses, and mammals included (respectively) as a predictive feature. Agreement between training constituent models with and without research efforts in mammalian and viral perspective was 99.7% [99.9% - 99.2%] (values in bracket are empirical confidence intervals derived from confidence intervals of constituent bootstrapped ensembles). Cohen's Kappa = 0.86 [0.85 - 0.89] (Kappa range: 0-1).

Results overview: Following majority voting, our divide and conquer approach suggested 21,327 (median, 90% CI = [2,926, 95,298]) unknown associations potentially exist between mammals and known viruses, (19,278 [2,566, 89,539] in wild or semi-domesticated mammals). These results indicate a ~4.37-fold increase ([~1.46, ~16.05]) in virus-mammal associations (~4.96 [~1.53, ~19.40] in wild and semi-domesticated mammals).

Example

The mammalian perspective: When including research effort into viruses in our mammalian perspective models, their results suggested a median of 161 [25, 437] unknown associations could form between WNV and terrestrial mammals (~3.88-fold increase [~1.45, ~8.80]). Similarly, the results indicated 66 [4, 331] new associations could form between our selected host (*R. leschenaultia*) and our viruses ~4.37-fold increase [~1.21, ~18.42]).

The viral perspective: When incorporating research effort into mammals into our viral perspective models, their results indicated a median of 67 [0, 197] new hosts of WNV ~2.2-fold increase [~1.00, ~4.52]). Results for our example host (*R. leschenaultia*) suggested 19 [3, 81] existing viruses could be found in this host (~2.00-fold increase [~1.16, ~5.26]).

The network perspective: same as manuscript.

Consolidation of perspectives: In the case of WNV, mammalian and viral perspectives achieved 88.16% agreement [97.91%, 73.82%]; mammals and network perspectives had 59.75% agreement [70.33%, 35.17%]; and viruses and network had 54.32% agreement [68.66%, 18.59%]. For of *R. leschenaultia*, these numbers were as follows: 95.89% [99.26%, 82.17%], 87.24% [95.04%, 76.37%], and 87.45% [95.04%, 75.84%], respectively.

The agreements between our perspectives across all possible associations were as follows: 98.02% [99.69%, 90.60%] between mammalian and viral perspectives, 96.75% [98.9%, 88.96%] between mammalian and network perspectives, and 97.09% [98.91%, 91.58%] between viral and network perspectives.

After voting: Our framework predicts a median of 189 [25, 504] new or undetected associations could be missing between WNV and terrestrial mammals (~4.38-fold increase [~1.48, ~10.00]) Similarly, our results indicated that *R. leschenaultia* could be susceptible to an additional 45 [5, 237] viruses not captured in our input (~3.37-fold increase [~1.26, ~13.47]). Supplementary Figure 7 illustrates top predicted and detected associations for WNV and *R. leschenaultia*.

Supplementary Figure 7 – Example showcasing final and intermediate predictions of West Nile Virus (WNV), and *Roussetus leschenaultii*, with models trained with research effort included in mammalian and viral perspectives. Panel A: Top 60 predicted mammalian species susceptible to WNV. Mammals were ordered by mean probability of predictions derived from mammalian (all models), viral (WNV models) and network perspectives, and top 60 were selected. Circles represent the following information in order: 1) whether the association is known (documented in our sources) or not (potential or undocumented). Hosts are omitted for known associations. 2) Mean probability of the three perspectives (per association). 3) Median mammalian perspective probabilities of predicted associations. These probabilities are obtained from 3,000 models (50 replicate models for each mammal), trained with viral features – SMOTE class balancing. 4) Median viral perspective probabilities of predicted associations (50 WNV replicate models trained with mammalian features – SMOTE class balancing). 5) Median network perspective probabilities of predicted associations (100 replicate models, balanced under-sampling). 6) Taxonomic order of predicted susceptible species. Orders are shortened as follows: Artiodactyla (Art), Carnivora (Crn), Chiroptera (Chp), primates (Prm), Rodentia (Rod), and Others (Oth). **Panel B: Top 50 predicted viruses of *R. leschenaultii*.** Viruses were ordered by mean probability of predictions derived from mammalian (*R. leschenaultii* models), viral (all models) and network perspectives. Circles as per Panel A. Baltimore represents Baltimore classification. **Panel C: Median probability of predicted WNV-mammal associations in each of the three perspectives per mammalian order.** Points represent susceptible species predicted by voting (at least two of the three perspectives – $n=137$). Median ensemble probability is computed in each perspective (50 replicate models for each virus/mammal, 100 replicate network models). Predictions derived from each perspective at 0.5 probability cut-off. **Panel D: Median probability of virus-*R. leschenaultii* associations in the three perspectives per Baltimore group.** Points represent susceptible species predicted by voting (at least two of the three perspectives – $n=64$), predictions are derived as per panel C.

Mammalian host range of viruses

Supplementary Figure 8 – Results (viruses) with research effort included in all perspectives. Panel A – Variable importance (relative contribution) of viral traits to mammalian perspective models. Variable importance is calculated for each constituent ensemble (n=699) of our mammalian perspective (median of a suite of 50 replicate models, trained with viral features, with SMOTE sampling), and then aggregated (mean) per each reported group (columns). **Panel B – Number of known and new mammalian species associated with each virus.** Rabies lyssavirus was excluded from panel B to allow for better visualisation. Top 40 (by number of new hosts) are labelled. Species in bold have over 150 predicted hosts (Supplementary Data 3 lists details of these viruses including CI). **Panel C – Predicted number of viruses per species of wild and semi-domesticated mammals (group by mammalian order).** Following orders (clockwise) are presented: Artiodactyla, Carnivora, Chiroptera, Perissodactyla, Primates, and Rodentia. Source of the silhouette graphics is PhyloPic.org. (Supplementary Data 4 lists aggregated results per mammalian order). Circles represent each mammalian species (with predicted viruses >0), coloured by number of known viruses previously not associated with this species. Boxplots indicate median (centre), the 25th and 75th percentiles (bounds of box) and inter quantile range (whiskers) and are aggregated at the order level. Large red circles with error bars (90% CI) illustrate the median number of known viruses per species in each order. Number of species presented (n) is as follows: All=1293 (Artiodactyla = 104, Carnivora = 177, Chiroptera = 548, Perissodactyla = 11, Primates = 171, and Rodentia = 282); Group I = 665 (94, 108, 157, 10, 159, 137); Group II = 392 (35, 121, 107, 1, 71, 57); Group III = 413 (86, 64, 121, 9, 47, 86); Group IV = 735 (97, 100, 220, 10, 148, 160); Group V = 1138 (92, 173, 530, 8, 108, 227); Group VI = 395 (69, 77, 40, 7, 141, 61); and Group VII = 178 (17, 6, 71, 2, 67, 15).

Relative importance of viral features: Supplementary Figure 8-A highlights relative importance of viral traits to our mammalian perspective models, with research effort of these viruses included as a predictive feature. Mean phylogenetic and ecological distances between potential and known hosts of focal virus had relative importance = 94.89% [73.8%, 100%], and 83.6% [42.66%, 100%],

respectively. Relative importance of maximum phylogenetic host breadth = 74.67% [16.59%, 100%].

Median relative influence of research effort (into each virus species included in our constituent models) across models trained in viral perspectives ($n=699$) was 63.45% [1.59%, 100%]. Supplementary Figure 14 visualises relative influence of mammalian research effort per each virus species (with two or more host species, $n=556$).

Mammalian host range (Supplementary Table 6, Supplementary Figure 8): Results of our framework suggest that the average mammalian host range of our viruses is 14.59 [4.88, 53.60] (average fold increase of ~ 3.12 [~ 1.23 , ~ 9.87] in number of hosts detected per virus). Overall, our approach indicate that the average host range of RNA viruses is 22.12 [7.07, 81.37] hosts (~ 3.89 -fold increase [~ 1.32 , ~ 13.50]). Whereas the average host range of DNA viruses is 7.92 [2.95, 29.03] hosts (~ 2.41 [~ 1.16 , ~ 6.89] fold increase).

Supplementary Table 6 –Predicted host-range of viruses per Baltimore group, family and transmission route.

Baltimore classification		Family		
	Predicted host range (~fold increase)			Predicted host range (~fold increase)
Group I	8.69 [3.17, 30.36] (~ 2.61 [~ 1.19 , ~ 6.81])	Bornaviridae	V	57.75 [13, 268.5] (~ 6.94 [~ 1.66 , ~ 37.01])
Group II	5.32 [2.22, 23.2] (~ 1.91 [~ 1.07 , ~ 6.07])	Orthomyxoviridae	V	69.75 [18.12, 199.25] (~ 8.44 [~ 1.59 , ~ 27.45])
Group III	28.89 [8.48, 93.7] (~ 4.13 [~ 1.4 , ~ 12.07])	Rhabdoviridae	V	49.25 [23.05, 143.89] (~ 6.48 [~ 1.67 , ~ 22.42])
Group IV	18.42 [5.32, 64.44] (~ 3.48 [~ 1.25 , ~ 10.61])	Hepeviridae	IV	83.67 [26.33, 222.33] (~ 6.83 [~ 2.33 , ~ 14.07])
Group V	24.59 [8.39, 97.39] (~ 4.12 [~ 1.34 , ~ 16.8])	Filoviridae	V	33.12 [8.12, 156.5] (~ 5.49 [~ 1.38 , ~ 24.69])
Group VI	28.77 [10.7, 97.23] (~ 5.24 [~ 1.58 , ~ 15.32])	Togaviridae	IV	48.85 [13.3, 155.5] (~ 5.61 [~ 1.67 , ~ 16.03])
Group V	29.86 [8, 118.29] (~ 2.8 [~ 1.28 , ~ 14.68])	Flaviviridae	IV	41.82 [11.14, 128.23] (~ 4.76 [~ 1.36 , ~ 15.41])
		Retroviridae	VI	28.77 [10.7, 97.23] (~ 5.24 [~ 1.58 , ~ 15.32])
Transmission route		Coronaviridae	IV	24.41 [6.41, 97.45] (~ 4.85 [~ 1.39 , ~ 18.13])
Direct	15.02 [5.21, 54.96] (~ 3.28 [~ 1.25 , ~ 10.22])	Poxviridae	I	24.27 [8.37, 85.1] (~ 4.96 [~ 1.56 , ~ 14.81])
Direct sexual	19.28 [6.47, 61.89] (~ 3.26 [~ 1.3 , ~ 9.28])	Reoviridae	III	34.55 [10.03, 112.03] (~ 4.62 [~ 1.49 , ~ 13.91])
Direct vertical	21.25 [7.06, 68.92] (~ 3.39 [~ 1.32 , ~ 10.18])	Paramyxoviridae	V	27.72 [8.74, 102] (~ 4.54 [~ 1.46 , ~ 14.18])
Indirect	20.86 [7.68, 71.19] (~ 3.39 [~ 1.3 , ~ 11.11])	Phenuiviridae	V	25.59 [7, 108.71] (~ 3.35 [~ 1.16 , ~ 15.26])
Ingestion	11.35 [4.22, 39.6] (~ 2.57 [~ 1.15 , ~ 7.62])	Peribunyaviridae	V	18.04 [5.83, 82.23] (~ 3.35 [~ 1.31 , ~ 16.47])
Inhalation	15.13 [4.72, 60.14] (~ 3.33 [~ 1.23 , ~ 11.56])	Hantaviridae	V	15.7 [4.96, 75.24] (~ 3.49 [~ 1.24 , ~ 16.18])
Environmental	21.86 [6.51, 83.11] (~ 3.87 [~ 1.3 , ~ 13.86])	Picornaviridae	IV	14.37 [4.45, 53.57] (~ 3.73 [~ 1.28 , ~ 10.68])
Vector	30.12 [8.53, 110.03] (~ 4.29 [~ 1.4 , ~ 16.19])	Pneumoviridae	V	33.33 [10.33, 117.89] (~ 3.95 [~ 1.32 , ~ 13.81])

Mammalian hosts of viruses

Supplementary Figure 9 – Results (Mammals). Panel A – variable importance (relative contribution) of mammalian traits to viral perspective models. Variable importance is calculated for each constituent model ($n=556$) of our viral perspective (trained with mammalian features), and then aggregated (median) per each reported group (columns). **Panel B – Number of known and new viruses associated with each mammal.** Labeled mammals are as follows: top 4 (by number of new viruses) for each of Artiodactyla, Carnivora, Chiroptera, Primates, Rodentia, and other orders. Species in bold have 100 or more predicted viruses (Supplementary Data 5). **Panel C – Top 18 genera (by number of predicted wild or semi-domesticated mammalian host species) in selected orders** (Other indicated results for all orders not included in the first five circles). Each order figure comprises the following circles (from outside to inside): 1) Number of hosts predicted to have an association with viruses within the viral genus. 2) Number of hosts detected to have association. 3) Number of hosts predicted to harbour viral zoonoses (i.e. known or predicted to share at least one virus species with humans). 4) Number of hosts predicted to share viruses with domesticated mammals of economic significance (domesticated mammals in orders: Artiodactyla, Carnivora, Lagomorpha and Perissodactyla). 5) Baltimore classification of the selected genera (Supplementary Data 6).

Relative importance of mammalian features: Supplementary Figure 9-A visualises relative importance of mammalian traits to our viral perspective models, with research effort of mammalian species included as a predictive feature. Our results suggest that distances to known hosts of viruses remained on average the top predictors of associations between the focal virus and our terrestrial mammals (taking all viral perspective models, $n=556$, into account). The breakdown was: 1) mean phylogenetic distance - all viruses = 98.48% [92.48%, 100%], DNA = 99.44% [95.6%, 100%] , RNA = 97.93% [90.64%, 100%] ; 2) mean ecological distance all viruses = 94.22% [71.69%,

100%], DNA = 96.35% [80.91%, 100%] , RNA = 92.98% [69.01%, 100%] . In addition, life-history traits significantly improved our models, in particular: longevity (all viruses = 60.74% [11.98%, 99.06%], DNA = 67.96% [11.21%, 99.69%], RNA = 57.12% [13.36%, 96.4%]); body mass (all viruses = 62.85% [5.17%, 97.65%], DNA = 71.95% [18.26%, 100%] , RNA = 57.33% [4.15%, 95.45%]), and reproductive traits (all viruses = 53.37% [5.64%, 96%], DNA = 59.39% [8.05%, 99.32%], RNA = 49.76% [4.89%, 92.1%]).

Median relative influence of research effort (into each mammalian species included in our constituent models) across models trained in viral perspectives ($n=556$) was 59.42% [1.15%, 99.79%]. The relative importance of mammalian research effort for DNA and RNA viruses = 65.76% [1.16%, 98.98%], and 57.62% [1.15%, 100%], respectively. Supplementary Figure 14 visualises relative influence of mammalian research effort per each virus species (with two or more host species, $n=556$).

Wild and semi-domesticated susceptible mammalian hosts of viruses (Supplementary Figure 9, Supplementary Table 7): results of our framework suggest a ~4.30-fold increase [~1.30, ~14.21] in the number of virus species that could potentially infect/associate with wild and or semi-domesticated mammals (17.16 [5.28, 67.1] viruses on average per host species). On average, wild or semi-domesticated hosts could be susceptible to additional 13.70 [1.82, 63.64] viruses on average (known mammalian viruses that are yet to be associated with these mammals).

Supplementary Table 7 – Predicted number of viruses per top 15 orders by fold increase in number of viruses predicted in wild or semi-domesticated mammalian hosts (per species). Values are derived per species and averaged per order.

Order/sub-order	Species	Average fold increase/species	Average virus range/species	Average new viruses/species
Tylopoda (part Artiodactyla)	6	~12.75 [~2.38, ~25.7]	31.5 [6.5, 105]	29 [4, 102.5]
Ruminantia (part Artiodactyla)	99	~9.27 [~1.89, ~22.79]	42.44 [11.34, 123.26]	36.95 [5.85, 117.77]
Primates	172	~6.88 [~1.59, ~20.17]	35.18 [11.14, 112.95]	28.45 [4.41, 106.22]
Suina (part Artiodactyla)	13	~8.55 [~1.4, ~26.34]	25.67 [4.58, 106.25]	22.67 [1.58, 103.25]
Perissodactyla	14	~6.13 [~1.84, ~16.2]	27.91 [9.09, 89.82]	23.36 [4.55, 85.27]
Cingulata	1	~10.33 [~2.33, ~37.25]	31 [7, 148]	28 [4, 145]
Lagomorpha	13	~5.48 [~1.6, ~15.28]	18.67 [4.58, 77.42]	15.75 [1.67, 74.5]
Rodentia	287	~4.78 [~1.29, ~18.66]	15.15 [3.89, 74.83]	12.52 [1.26, 72.21]
Carnivora	180	~3.89 [~1.35, ~13.65]	18.16 [6.25, 74.23]	13.93 [2.02, 70]
Hippopotamidae (part Artiodactyla)	2	~3.00 [~1.00, ~10.00]	3 [1, 19]	2 [0, 18]
Chiroptera	548	~2.85 [~1.13, ~9.35]	9.75 [3.24, 41.07]	7.32 [0.82, 38.65]
Scandentia	3	~2.95 [~1.18, ~10.31]	14.33 [6.33, 54.67]	9 [1, 49.33]
Didelphimorphia	5	~2.20 [~1, ~7.65]	4.8 [1.8, 24.2]	3 [0, 22.4]
Eulipotyphla	44	~2.06 [~1.05, ~12.99]	7.11 [2.86, 51.75]	4.57 [0.32, 49.2]
Pilosa	5	~1.54 [~1, ~14.96]	7.2 [4.6, 60]	2.6 [0, 55.4]

Supplementary Figure 10 – Number of previously undocumented (in EID2) viruses predicted per host species in relation to research effort for these hosts (without research effort as a predictive variable in models trained in the mammalian and viral perspectives). X-axis represents number of new viruses (previously unknown: undocumented or potential) predicted per host species via our divide and conquer approach (median probability cut-off ≥ 0.5 in each perspective, followed by majority voting). Y-axis represents research effort (logged) into the host species. Research effort is calculated as number of publications (as indexed in EID2) + number of genome sequences of the host species (for sequences, these are captured where the host species is the sequenced organism, or the host of the sequenced organism). Mammalian hosts are coloured per order. Humans and domestic species are labelled. Bold labels indicate ≥ 50 new viruses.

Supplementary Figure 11 – Number of previously undocumented (in EID2) viruses predicted per host species in relation to research effort for these hosts (with research effort as a predictive variable in models trained in the mammalian and viral perspectives). X-axis represents number of new viruses (previously unknown: undocumented or potential) predicted per host species via our divide and conquer approach (median probability cut-off ≥ 0.5 in each perspective, followed by majority voting). Y-axis represents research effort (logged) into the host species. Research effort is calculated as number of publications (as indexed in EID2) + number of genome sequences of the host species (for sequences, these are captured where the host species is the sequenced organism, or the host of the sequenced organism). Mammalian hosts are coloured per order. Humans and domestic species are labelled. Bold labels indicate ≥ 50 new viruses.

Supplementary Figure 12 – Number of previously undocumented (in EID2) mammalian hosts predicted per virus species in relation to research effort for these hosts (without research effort as a predictive variable in models trained in the mammalian and viral perspectives). X-axis represents number of new hosts (previously unknown: undocumented or potential) predicted per virus species via our divide and conquer approach (median probability cut-off ≥ 0.5 in each perspective, followed by majority voting). Y-axis represents research effort (logged) into the virus species. Research effort is calculated as number of publications (as indexed in EID2) + number of genome sequences of the virus species (the virus species, or any of its subspecies/strains, is the sequenced organism). Viruses are coloured per Baltimore classification. Viruses with total hosts species (known + predicted) ≥ 100 are labelled. Bold labels indicate ≥ 100 new hosts.

Supplementary Figure 13 – Number of previously undocumented (in EID2) mammalian hosts predicted per virus species in relation to research effort for these hosts (with research effort as a predictive variable in models trained in the mammalian and viral perspectives). X-axis represents number of new hosts (previously unknown: undocumented or potential) predicted per virus species via our divide and conquer approach (median probability cut-off ≥ 0.5 in each perspective, followed by majority voting). Y-axis represents research effort (logged) into the virus species. Research effort is calculated as number of publications (as indexed in EID2) + number of genome sequences of the virus species (the virus species, or any of its subspecies/strains, is the sequenced organism). Viruses are coloured per Baltimore classification. Viruses with total hosts species (known + predicted) ≥ 100 are labelled. Bold labels indicate ≥ 100 new hosts.

Supplementary Figure 14 – Log-log linear regression plots indicating association between research effort (logged) and number of new viruses or mammals predicted by our approach (logged). Without RE – indicate models trained without research effort incorporated into mammalian and viral perspectives. With RE – indicate models trained with research effort incorporated in all perspectives. R-squared and equation for each regression line are embedded in the figures. Spearman and Kendall correlations are embedded in titles. Shaded error bands represent 95% confidence intervals from the linear model.

Supplementary Figure 15 – Correlation between research effort, known associations (per species), and new associations (per species) predicted by voting and per each perspective. Without RE – indicate models trained without research effort incorporated into mammalian and viral perspectives.

Supplementary Figure 16 – Log-transformed scatter plots indicating association between fold increase and number of known viruses or mammals. Inset figures illustrate original variables, humans are removed from mammalian inset plots and Rabies is removed from viral inset plots for better visualisation. Without RE – indicate models trained without research effort incorporated into mammalian and viral perspectives. With RE – indicate models trained with research effort incorporated in all perspectives. Spearman and Kendall correlations are embedded in titles. X-axes (top) represent the number of viruses (logged) known to associate with included mammalian species (as per associations extracted from the EID2 database). Y-axes (top) represent fold-increases in number of viruses per mammalian species as predicted by our models. X-axes (bottom) represent the number of mammalian species (logged) known to associate with included viruses (as per associations extracted from the EID2 database). Y-axes (bottom) represent fold-increases in number of susceptible mammals per virus species as predicted by our models. Shaded error bands represent 95% confidence intervals from the linear model.

Supplementary Figure 17 – Variable importance of selected mammalian and viral top predictors (in terms of normalised variable importance) when research effort is included in constituent models trained in the mammalian and viral perspectives. Mammalian perspectives: X-axis represents host order. Following orders (clockwise) are presented: Artiodactyla, Carnivora, Chiroptera, Others (sloth silhouette), Perissodactyla, Primates, and Rodentia. Source of the silhouette graphics is PhyloPic.org. Y-axis represents normalised relative importance of the viral feature (% from 0 to 100). Points represent median relative influence/variable obtained from constituent mammalian models (50 replicate models per mammalian species with 2 or more known viruses). **Viral perspectives:** X-axis represents Baltimore classification. Y-axis represents normalised relative importance of the mammalian feature (% from 0 to 100). Points represent median relative influence/variable obtained from constituent viral models (50 replicate models per virus species with 2 or more known mammalian hosts).

Supplementary Results 2 – Test-set validation

Supplementary Table 8 – Differences in performance metrics between training and testing sets at full model and individual perspective levels. Values in square brackets indicate 90% confidence intervals.

These confidence intervals are derived from the confidence intervals of replicate models trained in each perspective (50 per mammalian/viral species, 100 per network perspective). In each perspective probability cut-off is set to 0.5.

Research effort included in network-perspective only		AUC	F1-Score	TSS	Sensitivity (Recall)	Specificity	Precision
Held-out test set	Vote (divide and conquer)	0.938 [0.862-0.959]	0.284 [0.464-0.124]	0.876 [0.724-0.918]	0.886 [0.728-0.949]	0.990 [0.997-0.969]	0.169 [0.34-0.067]
	Network perspective	0.929 [0.903-0.933]	0.104 [0.210-0.051]	0.859 [0.806-0.866]	0.895 [0.820-0.949]	0.964 [0.986-0.917]	0.055 [0.12-0.026]
	Mammalian perspective	0.890 [0.72-0.907]	0.115 [0.009-0.064]	0.781 [0.44-0.814]	0.809 [0.872-0.873]	0.971 [0.568-0.941]	0.062 [0.005-0.033]
	Viral perspective	0.839 [0.729-0.842]	0.181 [0.374-0.074]	0.677 [0.457-0.684]	0.691 [0.460-0.726]	0.986 [0.998-0.958]	0.104 [0.316-0.039]
Training	Vote (divide and conquer)	0.953 [0.929-0.961]	0.303 [0.464-0.156]	0.905 [0.859-0.922]	0.913 [0.862-0.942]	0.992 [0.996-0.980]	0.182 [0.317-0.085]
	Network perspective	0.920 [0.896-0.915]	0.058 [0.114-0.031]	0.839 [0.793-0.83]	0.897 [0.817-0.949]	0.942 [0.975-0.881]	0.030 [0.061-0.016]
	Mammalian perspective	0.914 [0.877-0.893]	0.178 [0.522-0.041]	0.827 [0.754-0.785]	0.842 [0.757-0.865]	0.985 [0.998-0.920]	0.100 [0.398-0.021]
	Viral perspective	0.865 [0.837-0.860]	0.249 [0.567-0.076]	0.731 [0.673-0.719]	0.739 [0.675-0.755]	0.992 [0.999-0.964]	0.150 [0.490-0.040]
Research effort included in all perspectives (network as above)		AUC	F1-Score	TSS	Sensitivity (Recall)	Specificity	Precision
Held-out test set	Vote (divide and conquer)	0.920 [0.823-0.944]	0.272 [0.526-0.093]	0.840 [0.646-0.888]	0.850 [0.648-0.931]	0.990 [0.998-0.958]	0.162 [0.442-0.049]
	Mammalian perspective	0.848 [0.725-0.865]	0.131 [0.284-0.035]	0.696 [0.449-0.73]	0.717 [0.453-0.837]	0.978 [0.996-0.893]	0.055 [0.12-0.026]
	Viral perspective	0.802 [0.717-0.831]	0.196 [0.373-0.067]	0.604 [0.433-0.663]	0.615 [0.436-0.708]	0.989 [0.998-0.955]	0.072 [0.206-0.018]
Training	Vote (divide and conquer)	0.952 [0.917-0.953]	0.298 [0.681-0.090]	0.904 [0.834-0.905]	0.912 [0.835-0.943]	0.992 [0.999-0.962]	0.117 [0.326-0.035]
	Mammalian perspective	0.918 [0.884-0.893]	0.163 [0.487-0.041]	0.835 [0.769-0.786]	0.852 [0.771-0.865]	0.983 [0.997-0.921]	0.03 [0.061-0.016]
	Viral perspective	0.865 [0.838-0.858]	0.230 [0.573-0.070]	0.730 [0.677-0.716]	0.739 [0.678-0.755]	0.991 [0.999-0.961]	0.090 [0.356-0.021]

Supplementary Table 9 – Validation results using agreement between two perspective. Predictions are considered positive if both perspectives predict them to be positive. Values in square brackets indicate 90% confidence intervals. These confidence intervals are derived from the confidence intervals of replicate models trained in each perspective (50 per mammalian/viral species, 100 per network perspective). In each perspective probability cut-off is set to 0.5.

Research effort included in network-perspective only		AUC	F1-Score	TSS	Sensitivity (Recall)	Specificity	Precision
Held-out test set	Network & viral	0.798 [0.712-0.844]	0.354 [0.471-0.173]	0.596 [0.423-0.689]	0.600 [0.424-0.704]	0.996 [0.999-0.985]	0.251 [0.531-0.099]
	Network & mammalian	0.868 [0.771-0.909]	0.309 [0.412-0.168]	0.736 [0.543-0.818]	0.743 [0.546-0.837]	0.993 [0.997-0.981]	0.195 [0.331-0.094]
	mammalian & viral	0.754 [0.642-0.812]	0.423 [0.386-0.22]	0.508 [0.283-0.624]	0.510 [0.284-0.634]	0.998 [0.999-0.999]	0.361 [0.605-0.133]
Training	Network & viral	0.841 [0.794-0.857]	0.406 [0.642-0.175]	0.683 [0.588-0.714]	0.686 [0.588-0.728]	0.997 [1.000-0.987]	0.289 [0.708-0.099]
	Network & mammalian	0.884 [0.823-0.901]	0.341 [0.65-0.115]	0.768 [0.645-0.802]	0.774 [0.646-0.826]	0.995 [0.999-0.975]	0.219 [0.655-0.062]
	mammalian & viral	0.817 [0.750-0.823]	0.501 [0.644-0.161]	0.634 [0.5-0.646]	0.636 [0.5-0.659]	0.998 [1.000-0.987]	0.413 [0.902-0.091]
Research effort included in all perspectives		AUC	F1-Score	TSS	Sensitivity (Recall)	Specificity	Precision
Held-out test set	Network & viral	0.788 [0.701-0.835]	0.331 [0.44-0.162]	0.577 [0.401-0.671]	0.581 [0.402-0.687]	0.995 [0.999-0.984]	0.231 [0.487-0.092]
	Network & mammalian	0.831 [0.706-0.888]	0.285 [0.442-0.114]	0.662 [0.412-0.776]	0.669 [0.413-0.805]	0.993 [0.999-0.971]	0.181 [0.476-0.061]
	mammalian & viral	0.715 [0.592-0.79]	0.368 [0.292-0.146]	0.429 [0.183-0.579]	0.431 [0.183-0.595]	0.998 [0.999-0.985]	0.321 [0.714-0.083]
Training	Network & viral	0.841 [0.795-0.857]	0.389 [0.638-0.165]	0.682 [0.589-0.714]	0.686 [0.590-0.728]	0.996 [0.999-0.986]	0.272 [0.696-0.093]
	Network & mammalian	0.883 [0.828-0.900]	0.336 [0.640-0.114]	0.767 [0.656-0.801]	0.772 [0.657-0.826]	0.994 [0.999-0.975]	0.214 [0.625-0.061]
	mammalian & viral	0.817 [0.759-0.823]	0.502 [0.662-0.16]	0.633 [0.519-0.645]	0.635 [0.519-0.658]	0.998 [0.999-0.987]	0.414 [0.915-0.091]

Supplementary Table 10 – Validation results using union of any two perspective. Predictions are considered positive if either of the included perspectives predict them to be positive. Values in square brackets indicate 90% confidence intervals. These confidence intervals are derived from the confidence intervals of replicate models trained in each perspective (50 per mammalian/viral species, 100 per network perspective). In each perspective probability cut-off is set to 0.5.

Research effort included in network-perspective only		AUC	F1-Score	TSS	Sensitivity (Recall)	Specificity	Precision
Held-out test set	Network & viral	0.946 [0.920-0.930]	0.095 [0.202-0.040]	0.892 [0.840-0.861]	0.934 [0.855-0.971]	0.958 [0.984-0.89]	0.050 [0.114-0.02]
	Network & mammalian	0.952 [0.945-0.931]	0.072 [0.12-0.036]	0.904 [0.891-0.862]	0.961 [0.923-0.985]	0.942 [0.968-0.877]	0.038 [0.064-0.018]
	mammalian & viral	0.951 [0.901-0.937]	0.107 [0.147-0.047]	0.902 [0.802-0.874]	0.938 [0.824-0.965]	0.964 [0.978-0.909]	0.057 [0.08-0.024]
Training	Network & viral	0.944 [0.939-0.917]	0.057 [0.121-0.027]	0.887 [0.878-0.835]	0.950 [0.904-0.976]	0.937 [0.974-0.858]	0.029 [0.065-0.013]
	Network & mammalian	0.954 [0.951-0.907]	0.054 [0.121-0.022]	0.907 [0.901-0.814]	0.975 [0.928-0.988]	0.932 [0.974-0.826]	0.028 [0.065-0.011]
	mammalian & viral	0.968 [0.964-0.929]	0.148 [0.5-0.036]	0.935 [0.927-0.859]	0.957 [0.931-0.961]	0.978 [0.996-0.897]	0.08 [0.342-0.018]
Research effort included in all perspectives		AUC	F1-Score	TSS	Sensitivity (Recall)	Specificity	Precision
Held-out test set	Network & viral	0.943 [0.919-0.929]	0.093 [0.205-0.039]	0.886 [0.838-0.857]	0.929 [0.853-0.970]	0.958 [0.985-0.887]	0.049 [0.116-0.02]
	Network & mammalian	0.946 [0.922-0.909]	0.08 [0.188-0.028]	0.893 [0.844-0.819]	0.943 [0.861-0.980]	0.949 [0.983-0.839]	0.042 [0.106-0.014]
	mammalian & viral	0.935 [0.85-0.906]	0.121 [0.33-0.031]	0.87 [0.7-0.813]	0.901 [0.706-0.950]	0.970d [0.994-0.863]	0.065 [0.215-0.016]
Training	Network & viral	0.943 [0.94-0.916]	0.056 [0.122-0.026]	0.886 [0.88-0.832]	0.950 [0.906-0.976]	0.937 [0.974-0.856]	0.029 [0.065-0.013]
	Network & mammalian	0.954 [0.953-0.908]	0.053 [0.12-0.022]	0.907 [0.905-0.815]	0.977 [0.932-0.988]	0.931 [0.973-0.827]	0.027 [0.064-0.011]
	mammalian & viral	0.966 [0.963-0.928]	0.133 [0.469-0.035]	0.932 [0.926-0.857]	0.956 [0.903-0.962]	0.976 [0.996-0.895]	0.072 [0.313-0.018]

Supplementary Table 11 – Validation results using mean probability of all perspective. Predictions are considered positive mean probability of the three perspective exceeds or equal a threshold (0.5). Values in square brackets indicate 90% confidence intervals. These confidence intervals are derived from the confidence intervals of replicate models trained in each perspective (50 per mammalian/viral species, 100 per network perspective). In each perspective probability cut-off is set to 0.5.

Research effort included in network-perspective only		AUC	F1-Score	TSS	Sensitivity (Recall)	Specificity	Precision
Held-out test set	Averaging	0.930 [0.853-0.953]	0.276 [0.481-0.098]	0.859 [0.705-0.901]	0.868 [0.708-0.933]	0.991 [0.997-0.973]	0.164 [0.364-0.052]
Training	Averaging	0.948 [0.923-0.956]	0.317 [0.487-0.141]	0.896 [0.847-0.912]	0.903 [0.850-0.93]	0.993 [0.997-0.983]	0.192 [0.342-0.076]
Research effort included in all perspectives		AUC	F1-Score	TSS	Sensitivity (Recall)	Specificity	Precision
Held-out test set	Averaging	0.914 [0.814-0.941]	0.265 [0.515-0.078]	0.828 [0.629-0.882]	0.837 [0.63-0.919]	0.991 [0.998-0.962]	0.157 [0.435-0.041]
Training	Averaging	0.947 [0.909-0.951]	0.294 [0.626-0.103]	0.894 [0.817-0.902]	0.901 [0.818-0.935]	0.993 [0.999-0.968]	0.176 [0.507-0.054]

Supplementary Results 3 – Systematic prediction of removed virus-mammal associations

We performed a systematic test to assess the ability of our framework to predict removed virus-mammal associations. The test was performed systematically by removing one documented association (known to exist) between one virus and one mammal. We factored in differences in host-ranges of viruses, and variations in number of viruses detected per mammalian species by adopting the following processes:

- Viruses:** We ordered virus species by the number of unique mammalian hosts detected (in EID2), and second to fourth letter of virus name. We selected the first virus in each count category and ordered their hosts by the second to fourth letter of host species name (i.e. excluding genus). We selected the first host (in order) and removed the resulting host-virus association. Supplementary Table 12 lists all removed associations from viruses' point of view.
- Mammals:** We ordered mammalian species by the number of unique virus species detected (in EID2), and second to fourth letter of species name (excluding genus). We selected the first mammalian species in each count category and ordered their viruses by the second to fourth letter of species name. We selected the first virus (in order) and removed the resulting host-virus association. Supplementary Table 13 lists all removed associations from mammals' point of view.

Following removal of each selected virus-mammal association, all dependent traits were re-calculated. We then retrained all constituent models (mammalian, viral, and network perspectives) using 10-fold cross validation and attempted to predict the removed link.

Supplementary Figure 18 – Results of systematic prediction of removed virus-mammal associations. Circles represent the following in order: whether the leave out test succeeded in predicting the removed interaction (black) or not (white); the mean probability across the three perspectives; probabilities derived from the mammalian perspective; probabilities derived from the viral perspective; probabilities derived from the network (motifs) perspective; the order of the host; and Baltimore classification of the virus. **Panel A** – results of viruses (interactions removed for each virus selected in Supplementary Table 12). **Panel B** – results of mammals (interactions removed for each mammal selected in Supplementary Table 13).

Supplementary Table 12 - Systematic prediction of removed virus-mammal associations (viruses): virus-mammal associations removed from viruses' point of view. No. is number of associations used to link with Supplementary Figure 18 (panel A – viruses), n = number of hosts per selected virus. Vote indicates if our framework predicted the removed association successfully (1) or not (0).

No.	N	mammal	virus	vote	No.	N	mammal	virus	vote
1	410	Homo sapiens	tai forest ebolavirus	1	23	23	Myodes glareolus	dobrava-belgrade orthohantavirus	1
2	134	Sus scrofa	akabane orthobunyavirus	1	24	22	Loxodonta africana	rabies lyssavirus	1
3	128	Bos taurus	new jersey vesiculovirus	1	25	21	Rousettus aegyptiacus	rabies lyssavirus	1
4	86	Pan troglodytes	macaca mulatta polyomavirus 1	1	26	20	Odocoileus hemionus	orf virus	1
5	73	Macaca mulatta	zika virus	1	27	19	Dama dama	rabies lyssavirus	0
6	71	Equus caballus	new jersey vesiculovirus	1	28	18	Saimiri sciureus	macaca mulatta polyomavirus 1	1
7	69	Ovis aries	akabane orthobunyavirus	1	29	17	Papio hamadryas	baboon orthoreovirus	1
8	65	Canis lupus familiaris	betacoronavirus 1	1	30	16	Eptesicus serotinus	bat astrovirus	1
9	54	Felis catus	rabies lyssavirus	1	31	15	Microtus agrestis	rabies lyssavirus	0
10	53	Rattus norvegicus	rat astrovirus	1	32	14	Syncerus caffer	rinderpest morbillivirus	1
11	47	Mus musculus	thailand orthohantavirus	1	33	13	Callithrix jacchus	rabies lyssavirus	0
12	45	Macaca fascicularis	tai forest ebolavirus	1	34	12	Chlorocebus sabaeus	pegivirus a	1
13	44	Eidolon helvum	rabies lyssavirus	1	35	11	Giraffa camelopardalis	betacoronavirus 1	1
14	40	Miniopterus schreibersii	bat astrovirus 1	1	36	10	Pteropus scapulatus	rabies lyssavirus	1
15	35	Chlorocebus aethiops	measles morbillivirus	1	37	9	Lontra canadensis	rabies lyssavirus	0
16	34	Zalophus californianus	sea lion mastadenovirus a	1	38	8	Camelus bactrianus	rotavirus a	1
17	33	Equus asinus	betacoronavirus 1	1	39	7	Macaca radiata	kyasanur forest disease virus	1
18	32	Cervus elaphus	rabies lyssavirus	1	40	6	Vulpes lagopus	rabies lyssavirus	1
19	30	Rattus rattus	rat astrovirus	1	41	5	Tylonycteris pachypus	bat circovirus	1
20	28	Rhinolophus ferrumequinum	bat bocavirus	1	42	4	Saguinus labiatus	hepatovirus a	1
21	26	Rangifer tarandus	orf virus	1	43	3	Plecotus rafinesquii	rabies lyssavirus	1
22	24	Capreolus capreolus	rotavirus a	1					

Supplementary Table 13 - Systematic prediction of removed virus-mammal associations (mammals): virus-mammal associations removed from mammals' point of view. No. is number of associations used to link with Supplementary Figure 18 (panel B – mammals), N = number of viruses per selected mammal.

Vote indicates if our framework predicted the removed association successfully (1) or not (0).

No.	N	mammal	virus	vote	No.	n	mammal	virus	vote
1	682	Chrysocyon brachyurus	rabies lyssavirus	1	24	26	Rattus tanezumi	rodent astrovirus	1
2	97	Callithrix jacchus	monkeypox virus	1	25	25	Cervus elaphus	respiratory syncytial virus	1
3	78	Chrysocyon brachyurus	canine morbillivirus	1	26	24	Equus caballus	saint louis encephalitis virus	1
4	62	Vulpes macrotis	carnivore protoparvovirus 1	1	27	23	Oligoryzomys flavescens	andes orthohantavirus	1
5	57	Lama glama	rotavirus a	1	28	22	Oligoryzomys flavescens	andes orthohantavirus	1
6	56	Equus caballus	west Nile virus	1	29	21	Tayassu pecari	suid alphaherpesvirus 1	0
7	55	Artibeus planirostris	influenza a virus	0	30	20	Boselaphus tragocamelus	small ruminant morbillivirus	1
8	50	Dama dama	tick-borne encephalitis virus	1	31	19	Acinonyx jubatus	felid alphaherpesvirus 1	1
9	48	Lama glama	pestivirus a	1	32	18	Pteropus giganteus	pegivirus b	0
10	45	Lama pacos	bluetongue virus	1	33	17	Macaca cyclopis	macaca mulatta polyomavirus 1	1
11	44	Pteropus scapulatus	japanese encephalitis virus	1	34	16	Equus caballus	deltapapillomavirus 4	1
12	43	Hipposideros caffer	bat paramyxovirus	1	35	14	Ovis dalli	orf virus	1
13	42	Equus caballus	ovine gammaherpesvirus 2	1	36	13	Chlorocebus sabaesus	pegivirus a	1
14	41	Pteropus scapulatus	dengue virus	1	37	12	Homo sapiens	dobrava-belgrade orthohantavirus	1
15	40	Equus caballus	eastern equine encephalitis virus	1	38	11	Equus caballus	akabane orthobunyavirus	1
16	39	Cynopterus brachyotis	bat coronavirus	1	39	10	Vulpes vulpes	rabbit hemorrhagic disease virus	0
17	38	Equus caballus	orthohepevirus a	1	40	9	Akodon azarae	argentinian mammarenavirus	1
18	36	Boselaphus tragocamelus	alcelaphine gammaherpesvirus 1	0	41	8	Macaca radiata	kyasanur forest disease virus	1
19	35	Saimiri sciureus	yellow fever virus	1	42	7	Mandrillus sphinx	macacine gammaherpesvirus 5	1
20	34	Cercopithecus nictitans	lymphocryptovirus 1	1	43	6	Homo sapiens	bhanja virus	1
21	32	Hipposideros abae	rift valley fever phlebovirus	1	44	5	Zalophus californianus	seal parapoxvirus	1
22	31	Puma yagouaroundi	alphacoronavirus 1	1	45	4	Rattus rattus	thailand orthohantavirus	1
23	30	Equus caballus	cowpox virus	1	46	3	Peromyscus leucopus	rabies lyssavirus	1

Supplementary Results 4 – The mammalian perspective

Results of our mammalian perspective suggested a median of 41,537 (90% CI [4,275, 238,971]) unknown associations could be missing from our original dataset. Our mammalian perspective models suggested an average 33.33 [7.39, 170.82] viruses per host (~8.3-fold increase [1.52, 51.04]), with average 28.93 [2.98, 166.41] new viruses per host.

When including research effort into viruses as a predictor into each constituent model trained in the mammalian perspective, our results suggested 40,612 [4,203, 227.119] unknown associations could be missing from our original dataset. On average there were 32.69 [7.34, 162.57] viruses per host (~8.07-fold increase [1.52, 525.5]), with average 28.28 [2.93, 158.16] new viruses per host.

Supplementary Figure 19 - Comparison of machine learning algorithms trained in viral feature space (mammalian perspective). Boxplots indicate median (centre), the 25th and 75th percentiles (bounds of box) and inter quantile range (whiskers) of performance metrics for 8 algorithms from 10-fold cross validation, trained with same training sets per mammal ($n=699$).

Supplementary Figure 20 - Selected classifiers trained in viral feature space (mammalian perspective). Boxplots indicate median (centre), the 25th and 75th percentiles (bounds of box) and inter quantile range (whiskers) of performance metrics from 50 runs with 10-fold cross validation of best performing tuned classifiers per mammal ($n=699$). Percentage of selected classifiers (number of times a classification algorithm was selected as best performing for a given mammalian species divided by number of times the algorithm was run in total) was as follows: avNNet=08.96%, GBM =10.90%, RF=11.64%, XGBoost=09.25%, SVM-RW=23.43%, SVM-LW=15.52%, SVM-P=11.19%, NB=09.10%.

Supplementary Figure 21 - Results of our mammalian perspective models. Panel A – Number of known and new mammalian species associated with each virus. Top 20 (by number of new hosts) are labelled. Species in bold have over 150 predicted hosts. Rabies lyssavirus was excluded from panel A to allow for better visualisation. **Panel B – Number of known and new viruses associated with each mammal.** Labelled mammals are as follows: top 4 (by number of new viruses) for each of Artiodactyla, Carnivora, Chiroptera, Primates, Rodentia, and other orders. Species in bold have 100 or more predicted viruses. **Panel C – Predicted number of viruses per each species of wild and semi-domesticated mammals (group by mammalian order).** Following orders (clockwise) are presented: Artiodactyla, Carnivora, Chiroptera, Perissodactyla, Primates, and Rodentia. Source of the silhouette graphics is PhyloPic.org. Circles represent each mammalian species (with predicted viruses >0), coloured by number of known viruses previously not associated with this species. Boxplots indicate median (centre), the 25th and 75th percentiles (bounds of box) and inter quantile range (whiskers) and are aggregated at the order level. Large red circles with error bars (90% CI) illustrate the median number of known viruses per species in each order. Number of species presented (n) is as follows: All=1293 (Artiodactyla = 104, Carnivora = 177, Chiroptera = 548, Perissodactyla = 11, Primates = 171, and Rodentia = 282); Group I = 656 (80, 105, 159, 9, 146, 157); Group II = 577 (72, 101, 155, 7, 118, 124); Group III = 584 (76,101,153,7,117,130); Group IV = 656 (81, 102, 178, 8, 131, 156); Group V = 1106 (81, 170, 522, 8, 118, 207); Group VI = 586 (74, 100, 153, 7, 127, 125); and Group VII = 575 (72,99,154,7,120,123). **Panel D – Top 18 genera (by number of predicted wild or semi-domesticated mammalian host species) in selected orders (Other indicated results for all orders not included in the first five circles).** Each order figure comprises the following circles (from outside to inside): 1) Number of hosts predicted to have an association with viruses within the viral genus. 2) Number of hosts detected to have association. 3) Number of hosts predicted to harbour viral zoonoses. 4) Number of hosts predicted to share viruses with domesticated mammals of economic significance (domesticated mammals in orders: Artiodactyla, Carnivora, Lagomorpha and Perissodactyla). 5) Baltimore classification.

Supplementary Figure 22 - Results of our mammalian perspective models with inclusion of research effort into viruses as a predictor in constituent models. Panel A – Number of known and new mammalian species associated with each virus. Top 20 (by number of new hosts) are labelled. Species in bold have over 150 predicted hosts. Rabies lyssavirus was excluded from panel A to allow for better visualisation. Panel B – Number of known and new viruses associated with each mammal. Labelled mammals are as follows: top 4 (by number of new viruses) for each of Artiodactyla, Carnivora, Chiroptera, Primates, Rodentia, and other orders. Species in bold have 100 or more predicted viruses. Panel C – Predicted number of viruses per each species of wild and semi-domesticated mammals (group by mammalian order). Following orders (clockwise) are presented: Artiodactyla, Carnivora, Chiroptera, Perissodactyla, Primates, and Rodentia. Source of the silhouette graphics is PhyloPic.org. Circles represent each mammalian species (with predicted viruses >0), coloured by number of known viruses previously not associated with this species. Boxplots represent aggregation at the order level. Large red circles with error bars (90% CI) illustrate the median number of known viruses per species in each order. Number of species presented (n) is as follows: All=1293 (Artiodactyla = 104, Carnivora = 177, Chiroptera = 548, Perissodactyla = 11, Primates = 171, and Rodentia = 282); Group I = 656 (60, 105, 159, 9, 146, 157); Group II = 576 (72, 100, 155, 7, 118, 124); Group III = 581 (76,98, 153,7, 117,130); Group IV = 653 (81, 99, 178, 8, 131, 156); Group V = 1106 (81, 170, 522, 8, 118, 207); Group VI = 584 (74, 98, 153, 7, 127, 125); and Group VII = 573 (72, 97, 154,7,120,123). Panel D – Top 18 genera (by number of predicted wild or semi-domesticated mammalian host species) in selected orders (Other indicated results for all orders not included in the first five circles). Each order figure comprises the following circles (from outside to inside): 1) Number of hosts predicted to have an association with viruses within the viral genus. 2) Number of hosts detected to have association. 3) Number of hosts predicted to harbour viral zoonoses. 4) Number of hosts predicted to share viruses with domesticated mammals of economic significance (domesticated mammals in orders: Artiodactyla, Carnivora, Lagomorpha and Perissodactyla). 5) Baltimore classification.

Supplementary Results 5 – the viral perspective

When using mammalian features to predict potential associations with multi-host viruses, the results of our selected models indicated that 21,352 (median, 90% CI [2,536, 95,630]) unknown associations could be missing from the original bipartite network. Our mammalian perspective models suggested an average 15.10 [4.84, 55.63] hosts per virus (~3.74-fold increase [~1.26, ~16.43]), with average 11.65 [1.38, 52.17] new hosts per virus.

When incorporating research effort into mammalian species as a predictor into each constituent model trained in the viral perspective, our viral perspective suggested median = 20,800 [2,366, 94,352] unknown associations could form between our viruses and mammalian hosts. indicated an average 14.8 [4.74, 54.93] hosts per virus (~3.61-fold increase [~1.20, ~16.18]), with average 11.35 [1.29, 51.47] new hosts per virus.

Supplementary Figure 23 - Comparison of machine learning algorithms trained in mammalian feature space (viral perspective). Boxplots indicate median (centre), the 25th and 75th percentiles (bounds of box) and inter quantile range (whiskers) of performance metrics for 8 algorithms from 10-fold cross validation, each trained with same training sets per each virus ($n=556$).

Supplementary Figure 24 - Selected classifiers trained in mammalian feature space (viral perspective). Boxplots indicate median (centre), the 25th and 75th percentiles (bounds of box) and inter quantile range (whiskers) of performance metrics for 8 algorithms from 50 runs with 10-fold cross validation of best performing tuned classifiers per each virus ($n=556$). Percentage of selected classifiers (number of times a classification algorithm was selected as best performing for a given virus species divided by number of times the algorithm was run in total) was as follows: avNNet =6.18%, GBM =16.6%, RF=13.13%, XGBoost=20.46%, SVM-RW=14.48%, SVM-LW=4.63%, SVM-P=5.02%, NB=19.5%.

Supplementary Figure 25 - results of our viral perspective models. Panel A – Number of known and new mammalian species associated with each virus. Top 20 (by number of new hosts) are labelled. Species in bold have over 150 predicted hosts. Rabies lyssavirus was excluded from panel A to allow for better visualisation. **Panel B – Number of known and new viruses associated with each mammal.** Labelled mammals are as follows: top 4 (by number of new viruses) for each of Artiodactyla, Carnivora, Chiroptera, Primates, Rodentia, and other orders. Species in bold have 100 or more predicted viruses. **Panel C – Predicted number of viruses per each species of wild and semi-domesticated mammals (group by mammalian order).** Following orders (clockwise) are presented: Artiodactyla, Carnivora, Chiroptera, Perissodactyla, Primates, and Rodentia. Source of the silhouette graphics is PhyloPic.org. Circles represent each mammalian species (with predicted viruses >0), coloured by number of known viruses previously not associated with this species. Boxplots indicate median (centre), the 25th and 75th percentiles (bounds of box) and inter quantile range (whiskers) and are aggregated at the order level. Large red circles with error bars (90% CI) illustrate the median number of known viruses per species in each order. Number of species presented (*n*) is as follows: All=1293 (Artiodactyla = 104, Carnivora = 177, Chiroptera = 548, Perissodactyla = 11, Primates = 171, and Rodentia = 282); Group I = 1156 (102, 169, 438, 11, 164, 272); Group II = 1135 (99, 170, 437, 11, 147, 271); Group III = 1127 (99, 165, 435, 11, 147, 270); Group IV = 1139 (100, 165, 440, 11, 152, 271); Group V = 1255 (99, 174, 545, 11, 148, 278); Group VI = 1136 (100, 165, 435, 11, 155, 270); and Group VII = 1126 (99, 164, 435, 11, 147, 270). **Panel D – Top 18 genera (by number of predicted wild or semi-domesticated mammalian host species) in selected orders (Other indicated results for all orders not included in the first five circles).** Each order figure comprises the following circles (from outside to inside): 1) Number of hosts predicted to have an association with viruses within the viral genus. 2) Number of hosts detected to have association. 3) Number of hosts predicted to harbour viral zoonoses. 4) Number of hosts predicted to share viruses with domesticated mammals of economic significance (domesticated mammals in orders: Artiodactyla, Carnivora, Lagomorpha and Perissodactyla). 5) Baltimore classification.

Supplementary Figure 26 - results of our viral perspective models with inclusion of mammalian research effort as a predictor in constituent models. Panel A – Number of known and new mammalian species associated with each virus. Top 20 (by number of new hosts) are labelled. Species in bold have over 150 predicted hosts. Rabies lyssavirus was excluded from panel A to allow for better visualisation. **Panel B – Number of known and new viruses associated with each mammal.** Labelled mammals are as follows: top 4 (by number of new viruses) for each of Artiodactyla, Carnivora, Chiroptera, Primates, Rodentia, and other orders. Species in bold have 100 or more predicted viruses. **Panel C – Predicted number of viruses per each species of wild and semi-domesticated mammals (group by mammalian order).** Following orders (clockwise) are presented: Artiodactyla, Carnivora, Chiroptera, Perissodactyla, Primates, and Rodentia. Source of the silhouette graphics is PhyloPic.org. Circles represent each mammalian species (with predicted viruses >0), coloured by number of known viruses previously not associated with this species. Boxplots indicate median (centre), the 25th and 75th percentiles (bounds of box) and inter quantile range (whiskers) and are aggregated at the order level. Large red circles with error bars (90% CI) illustrate the median number of known viruses per species in each order. Number of species presented (*n*) is as follows: All=1293 (Artiodactyla = 104, Carnivora = 177, Chiroptera = 548, Perissodactyla = 11, Primates = 171, and Rodentia = 282); Group I = 1156 (102, 169, 438, 11, 164, 272); Group II = 1135 (99, 170, 437, 11, 147, 271); Group III = 1127 (99, 165, 435, 11, 147, 270); Group IV = 1139 (100, 165, 440, 11, 152, 271); Group V = 1255 (99, 174, 545, 11, 148, 278); Group VI = 1136 (100, 165, 435, 11, 155, 27); and Group VII = 1126 (99, 164, 435, 11, 147, 270). **Panel D – Top 18 genera (by number of predicted wild or semi-domesticated mammalian host species) in selected orders (Other indicated results for all orders not included in the first five circles).** Each order figure comprises the following circles (from outside to inside): 1) Number of hosts predicted to have an association with viruses within the viral genus. 2) Number of hosts detected to have association. 3) Number of hosts predicted to harbour viral zoonoses. 4) Number of hosts predicted to share viruses with domesticated mammals of economic significance (domesticated mammals in orders: Artiodactyla, Carnivora, Lagomorpha and Perissodactyla). 5) Baltimore classification.

Supplementary Results 6 – The network perspective

models suggested 76,081 (median, 90% CI [27,738, 205,814]) unknown associations could be missing from the original bipartite network. Our network perspective models indicated an average range = 57.39 [23.72, 147.73] viruses per mammalian host (~14.19-fold increase [~4.78, ~57.23]), with 52.98 [19.32, 143.32] new viruses per host. Conversely, they suggested an average range= 44.93 [18.58, 114.6] mammalian hosts per virus (~11.74-fold increase [~3.69, ~41.74]), with average 41.47 [15.13, 111.14] new hosts per virus.

Supplementary Figure 27 - Comparison of machine learning algorithms trained in mammalian feature space (network perspective). Boxplots indicate median (centre), the 25th and 75th percentiles (bounds of box) and inter quantile range (whiskers) of performance metrics for 8 algorithms from 10-fold cross validation (100 replicate models, trained with balanced sample of 2,000 associations: 1,000 known and 1,000 unknown), results derived from applying each trained model to the set of possible mammal virus interactions ($n=2,722,656$).

Supplementary Figure 28 - results of our network perspective models. Panel A – Number of known and new mammalian species associated with each virus. Top 30 (by number of new hosts) are labelled. Species in bold have over 150 predicted hosts. Rabies lyssavirus was excluded from panel A to allow for better visualisation. **Panel B – Number of known and new viruses associated with each mammal.** Labelled mammals are as follows: top 4 (by number of new viruses) for each of Artiodactyla, Carnivora, Chiroptera, Primates, Rodentia, and other orders. Species in bold have 100 or more predicted viruses. **Panel C – Predicted number of viruses per each species of wild and semi-domesticated mammals (group by mammalian order).** Following orders (clockwise) are presented: Artiodactyla, Carnivora, Chiroptera, Perissodactyla, Primates, and Rodentia. Source of the silhouette graphics is PhyloPic.org. Circles represent each mammalian species (with predicted viruses >0), coloured by number of known viruses previously not associated with this species. Boxplots indicate median (centre), the 25th and 75th percentiles (bounds of box) and inter quantile range (whiskers) and are aggregated at the order level. Large red circles with error bars (90% CI) illustrate the median number of known viruses per species in each order. Number of species presented (*n*) is as follows: All=1293 (Artiodactyla = 104, Carnivora = 177, Chiroptera = 548, Perissodactyla = 11, Primates = 171, and Rodentia = 282); Group I = 1293 (104, 177, 548, 11, 171, 282); Group II = 1293 (104, 177, 548, 11, 171, 282); Group III = 1293 (104, 177, 548, 11, 171, 282); Group IV = 1293 (104, 177, 548, 11, 171, 282); Group V = 1293 (104, 177, 548, 11, 171, 282); Group VI = 1293 (104, 177, 548, 11, 171, 282); and Group VII = 1293 (104, 177, 548, 11, 171, 282). **Panel D – Top 18 genera (by number of predicted wild or semi-domesticated mammalian host species) in selected orders (Other indicated results for all orders not included in the first five circles).** Each order figure comprises the following circles (from outside to inside): 1) Number of hosts predicted to have an association with viruses within the viral genus. 2) Number of hosts detected to have association. 3) Number of hosts predicted to harbour viral zoonoses. 4) Number of hosts predicted to share viruses with domesticated mammals of economic significance (domesticated mammals in orders: Artiodactyla, Carnivora, Lagomorpha and Perissodactyla). 5) Baltimore classification.

Supplementary Figure 29 - Partial dependence plots of each of our network features (motif-features and research effort) in our network perspective. X-axes show predictor value (logged – number of motifs in which the association (response) is the focal-association, research effort into the mammal and the virus). Y-axes show the effect on the probability of the virus-mammal association (0 to 1) when changing only the value of given predictor while keeping all other predictors constant. Individual lines show the partial dependence per each run (replicate model – $n = 100$, each trained with balanced sample comprising 1,000 known positive and 1,000 unknown associations) of the ensemble. The smoothed lines (smoothed conditional means) illustrate the overall trend of partial dependence between our response variable and each of our network features. Partial dependence measures the response for an individual variable in a machine-learning model (here SVM-RW), while holding all other variable constant. Partial dependence plots visualise the potentially non-linear relationships between each predictor in our network perspective and the response variable (whether a given mammal could be susceptible to a given virus).

Supplementary Results 7 – Additional Results

Viruses found in humans: Our multi-perspective framework generates predictions for each potential virus-mammal association ($n=2,722,656$ between 1,896 viruses and 1,436 terrestrial mammals). Here we highlight results for humans. In addition to 425 virus species known to affect humans, our model predicted 16 undocumented (in EID2) virus species that could potentially be found in humans. Supplementary Table 9 presents these viruses.

Supplementary Table 14 – Predicted viruses that could potentially be found in humans (not documented in input dataset). Baltimore is Baltimore classification (7 groups I to VII). *Mammalian* lists probabilities drawn from *median* mammalian perspective (Homo sapiens model trained with viral traits). *Viral* lists median probabilities drawn for humans from viral perspective (16 virus models trained with mammalian traits). *Network* lists probabilities drawn from network perspective (trained with motif features). *Mean*: is average probability across the three perspectives. Voting was drawn for each perspective (1 if probability 0.5).

Virus species	Baltimore	Mammalian	Viral	Network	Mean
murid betaherpesvirus 1	I	0.003	0.541	0.999	0.514
cercopithecine alphaherpesvirus 9	I	0.042	0.781	0.999	0.607
lymphocryptovirus 1	I	0.073	1.000	0.999	0.691
deltapapillomavirus 4	I	0.002	0.994	0.997	0.665
bat rotavirus	III	0.005	0.601	0.988	0.531
bat hepatovirus	IV	0.006	0.979	0.973	0.653
primate astrovirus	IV	0.045	0.618	0.999	0.554
enterovirus j	IV	0.158	0.707	0.999	0.621
enterovirus f	IV	0.007	0.708	0.996	0.570
mamastrovirus 5	IV	0.574	0.224	0.998	0.599
aravan lyssavirus	V	0.004	0.519	0.955	0.493
beilong jeilongvirus	V	0.001	0.518	0.990	0.503
coastal plains tibrovirus	V	0.008	0.627	0.996	0.544
influenza d virus	V	0.019	0.902	0.998	0.640
simian immunodeficiency virus	VI	0.459	0.696	0.999	0.718
simian retrovirus 5	VI	0.021	0.575	0.997	0.531

Species Richness: Overall, we found a negative correlation between number of included mammalian species per taxonomic order, and the number of new viruses predicted per species. Spearman= -0.34, Kendall=-0.27 (when taking only wild and semi-domesticated mammals into consideration: Spearman = -0.32 - Kendall = -0.25). Similar correlation results were obtained when research effort was included in all perspectives.

Supplementary Results 8 – Sequence-evidence pipeline

We trained an independent pipeline including only the 3,534 associations supported by evidence extracted from meta-data accompanying nucleotide sequences, as indexed in EID2 (55.82% of associations which entered our original models). When using only this evidence, the average host range of included viruses is 1.21, and average number of viruses per wild or semi-domesticated mammalian species is 1.63.

Results overview: Following majority voting, sequence-evidence pipeline indicated that 15,721 (median, 90% CI = [1,603, 88,553]) unknown associations could potentially exist (13,930 [1,298, 83,043] in wild or semi-domesticated mammals) – equivalent to a \sim 5.78-fold increase ([\sim 1.49, \sim 28.02]) in sequence-evidence supported virus-mammal associations (\sim 7.07 [\sim 1.57, \sim 37.18] in similarly supported wild and semi-domesticated mammals).

Performance metrics estimated using training-data were as follows: AUC=0.959 [0.922-0.957], F1-score=0.257 [0.724-0.066], and TSS=0.926 [0.845, 0.915].

Mammalian host range of viruses: Results of this pipeline indicated that the average mammalian host range of included viruses is 8.58 [1.90, 45.1] (average fold increase of \sim 3.37 [\sim 1.17, \sim 19.21] in number of susceptible species predicted per virus). Overall, our approach indicate that the average host range of RNA viruses is 12.98 [2.65, 69.64] susceptible species (\sim 4.51- fold increase [\sim 1.28, \sim 27.69]). Whereas the average host range of DNA viruses is 4.67 [1.23, 23.36] susceptible species (\sim 2.37 [\sim 1.10, \sim 11.70] fold increase).

Mammalian hosts of viruses: Results of this pipeline suggest a \sim 4.05-fold increase [\sim 1.21, \sim 21.46] in the number of virus species that could potentially infect/associate with wild and or semi-domesticated mammals (17.16 [2.55, 60.65] viruses on average per host species). On average, wild or semi-domesticated mammals could be susceptible to additional 9.90 [0.92, 59.02] viruses on average (known mammalian viruses that are yet to be associated with these mammals).

Supplementary Figure 30 – Density heatmaps illustrating agreement of probabilities derived in each perspective when using both sources of evidence and when using sequence-evidence only. Axes illustrate probability derived (in each perspective) using either sequence-evidence, or both sources (original divide-and-conquer as reported in the main text). Presented predictions are limited to 34,354 associations either known or predicted (using voting with median probability cut-off \geq 0.5) by at least one of the two pipelines (27,163 when using both source and 18,998 when using sequence-evidence).

Supplementary Figure 31 – Density heatmaps illustrating pair-wise agreement between perspectives, when using both sources of evidence and when using sequence-evidence only. Axes illustrate probability derived from each perspective, when using either sequence-evidence, publication-evidence or both sources (original divide-and-conquer as reported in the main text). Presented predictions are limited, in each panel, as follows: the 27,163 known or predicted when using both source, and 18,998 known or predicted when using sequence-evidence.

Supplementary References

1. Federhen, S. The NCBI Taxonomy database. *Nucleic Acids Res.* **40**, D136-43 (2012).
2. Wardeh, M., Risley, C., McIntyre, M. K., Setzkorn, C. & Baylis, M. Database of host-pathogen and related species interactions, and their global distribution. *Sci. Data* **2**, (2015).
3. Sanjuán, R. *et al.* Viral Mutation Rates Viral Mutation Rates □. *J. Virol.* **84**, 9733–9748 (2010).
4. Coffin, J. M. Structure and Classification of Retroviruses. in *The Retroviridae* 19–49 (Springer US, 1992). doi:10.1007/978-1-4615-3372-6_2.
5. Nisole, S. & Saïb, A. Early steps of retrovirus replicative cycle. *Retrovirology* vol. 1 (2004).
6. Hulo, C. *et al.* ViralZone: A knowledge resource to understand virus diversity. *Nucleic Acids Res.* **39**, D576 (2011).
7. Wawrzyniak, P., Plucienniczak, G. & Bartosik, D. The different faces of rolling-circle replication and its multifunctional initiator proteins. *Frontiers in Microbiology* vol. 8 (2017).
8. Lin, X. *et al.* Order and disorder control the functional rearrangement of influenza hemagglutinin. *Proc. Natl. Acad. Sci. U. S. A.* **111**, 12049–54 (2014).
9. Sicard, A., Michalakis, Y., Gutiérrez, S. & Blanc, S. The Strange Lifestyle of Multipartite Viruses. *PLoS Pathogens* vol. 12 (2016).
10. Rey, F. A. & Lok, S. M. Common Features of Enveloped Viruses and Implications for Immunogen Design for Next-Generation Vaccines. *Cell* vol. 172 1319–1334 (2018).
11. Benson, D. A. *et al.* GenBank. *Nucleic Acids Res.* **41**, D36-42 (2013).
12. Bethesda (MD): National Library of Medicine (US), N. C. for B. I. GenBank [Internet]. <https://www.ncbi.nlm.nih.gov/nucleotide/> (1982).
13. Yakovchuk, P., Protozanova, E. & Frank-Kamenetskii, M. D. Base-stacking and base-pairing

- contributions into thermal stability of the DNA double helix. Yakovchuk, P., Protozanova, E. & Frank-Kamenetskii, M. D. Base-stacking and base-pairing contributions into thermal stability of the DNA double helix. *Nucleic Acids Res.* **34**, 564–574 (2006).
14. Komarova, N. L. Viral reproductive strategies: How can lytic viruses be evolutionarily competitive? *J. Theor. Biol.* **249**, 766–84 (2007).
 15. Lefkowitz, E. J. *et al.* Virus taxonomy: The database of the International Committee on Taxonomy of Viruses (ICTV). *Nucleic Acids Res.* **46**, D708–D717 (2018).
 16. Woolhouse, M. E. J. & Brierley, L. Epidemiological characteristics of human-infective RNA viruses. *Sci. Data* **5**, (2018).
 17. Olival, K. J. *et al.* Host and viral traits predict zoonotic spillover from mammals. *Nature* **546**, 646–650 (2017).
 18. Guth, S., Visher, E., Boots, M. & Brook, C. E. Host phylogenetic distance drives trends in virus virulence and transmissibility across the animal–human interface. *Philos. Trans. R. Soc. B Biol. Sci.* **374**, 20190296 (2019).
 19. Longdon, B., Brockhurst, M. A., Russell, C. A., Welch, J. J. & Jiggins, F. M. The Evolution and Genetics of Virus Host Shifts. *PLoS Pathog.* **10**, e1004395 (2014).
 20. Fritz, S. A., Bininda-Emonds, O. R. P. & Purvis, A. Geographical variation in predictors of mammalian extinction risk: big is bad, but only in the tropics. *Ecol. Lett.* **12**, 538–549 (2009).
 21. Isaac, N. J. B., Turvey, S. T., Collen, B., Waterman, C. & Baillie, J. E. M. Mammals on the EDGE: Conservation Priorities Based on Threat and Phylogeny. *PLoS One* **2**, e296 (2007).
 22. Kembel, S. W. *et al.* Picante: R tools for integrating phylogenies and ecology. *Bioinformatics* **26**, 1463–1464 (2010).
 23. Vane-Wright, R. I., Humphries, C. J. & Williams, P. H. What to protect?—Systematics and the agony of choice. *Biol. Conserv.* **55**, 235–254 (1991).
 24. Park, A. W. *et al.* Characterizing the phylogenetic specialism–generalism spectrum of mammal parasites. *Proc. R. Soc. B Biol. Sci.* **285**, 20172613 (2018).
 25. Jones, K. E. *et al.* PanTHERIA: a species-level database of life history, ecology, and geography of extant and recently extinct mammals. *Ecology* **90**, 2648–2648 (2009).
 26. Gnanadesikan, G. E., Pearse, W. D. & Shaw, A. K. Evolution of mammalian migrations for refuge, breeding, and food. *Ecol. Evol.* **7**, 5891–5900 (2017).
 27. Wilman, H. *et al.* EltonTraits 1.0: Species-level foraging attributes of the world’s birds and mammals. *Ecology* **95**, 2027–2027 (2014).
 28. IUCN 2018. The IUCN Red List of Threatened Species. Version 2018-2. <http://www.iucnredlist.org>.
 29. DE MAGALHÃES, J. P. & COSTA, J. A database of vertebrate longevity records and their relation to other life-history traits. *J. Evol. Biol.* **22**, 1770–1774 (2009).
 30. Gower, J. C. A General Coefficient of Similarity and Some of Its Properties. *Biometrics* **27**, 857 (1971).
 31. Pavoine, S., Vallet, J., Dufour, A.-B., Gachet, S. & Daniel, H. On the challenge of treating various types of variables: application for improving the measurement of functional diversity. *Oikos* **118**, 391–402 (2009).
 32. McIntyre, K. M. *et al.* Systematic Assessment of the Climate Sensitivity of Important Human and Domestic Animals Pathogens in Europe. *Sci. Rep.* **7**, 7134 (2017).
 33. Hay, S. I. *et al.* Global mapping of infectious disease. *Philos. Trans. R. Soc. Lond. B. Biol. Sci.* **368**, 20120250 (2013).
 34. Anyamba, A. *et al.* Global Disease Outbreaks Associated with the 2015–2016 El Niño Event. *Sci. Rep.* **9**, 1930 (2019).
 35. Jones, A. E. *et al.* Bluetongue risk under future climates. *Nat. Clim. Chang.* **9**, 153–157 (2019).
 36. Caminade, C., McIntyre, K. M. & Jones, A. E. Impact of recent and future climate change on vector-borne diseases. *Ann. N. Y. Acad. Sci.* **1436**, 157–173 (2019).
 37. Karesh, W. B. *et al.* Ecology of zoonoses: natural and unnatural histories. *Lancet* **380**, 1936–1945 (2012).
 38. Hassell, J. M., Begon, M., Ward, M. J. & Fèvre, E. M. Urbanization and Disease Emergence: Dynamics at the Wildlife-Livestock-Human Interface. *Trends Ecol. Evol.* **32**, 55–67 (2017).
 39. Gilbert, M. *et al.* Global distribution data for cattle, buffaloes, horses, sheep, goats, pigs, chickens and ducks in 2010. *Sci. Data* **5**, 180227 (2018).
 40. University, C. for I. E. S. I. N.-C.-C. Gridded Population of the World, Version 4 (GPWv4):

- Population Density Adjusted to Match 2015 Revision UN WPP Country Totals, Revision 10. (2017).
41. Harris, I., Jones, P. D., Osborn, T. J. & Lister, D. H. Updated high-resolution grids of monthly climatic observations - the CRU TS3.10 Dataset. *Int. J. Climatol.* **34**, 623–642 (2014).
 42. IUCN, I. U. for C. of N.- & University, C. for I. E. S. I. N.-C.-C. Gridded Species Distribution: Global Mammal Richness Grids, 2015 Release. (2015).
 43. Hopkins, M. E. & Nunn, C. L. A global gap analysis of infectious agents in wild primates. *Divers. Distrib.* **13**, 561–572 (2007).
 44. Lloyd-Smith, J. O. *et al.* Epidemic Dynamics at the Human-Animal Interface. *Science (80-.).* **326**, 1362–1367 (2009).
 45. Robinson, T. P. *et al.* Mapping the Global Distribution of Livestock. *PLoS One* **9**, e96084 (2014).
 46. Jenkins, C. N., Pimm, S. L. & Joppa, L. N. Global patterns of terrestrial vertebrate diversity and conservation. *Proc. Natl. Acad. Sci. U. S. A.* **110**, E2602-10 (2013).
 47. Jones, B. A. *et al.* Zoonosis emergence linked to agricultural intensification and environmental change. *Proc. Natl. Acad. Sci.* **110**, 8399–8404 (2013).
 48. Weaver, S. C. Urbanization and geographic expansion of zoonotic arboviral diseases: mechanisms and potential strategies for prevention. *Trends Microbiol.* **21**, 360–3 (2013).
 49. Eskew, E. A. & Olival, K. J. De-urbanization and Zoonotic Disease Risk. *Ecohealth* (2018) doi:10.1007/s10393-018-1359-9.
 50. Jones, K. E. *et al.* Global trends in emerging infectious diseases. *Nature* **451**, 990–993 (2008).
 51. Dunn, R. R., Davies, T. J., Harris, N. C. & Gavin, M. C. Global drivers of human pathogen richness and prevalence. *Proc. R. Soc. B Biol. Sci.* **277**, 2587–2595 (2010).
 52. Wolfe, N. D., Dunavan, C. P. & Diamond, J. Origins of major human infectious diseases. *Nature* **447**, 279–283 (2007).
 53. Allen, T. *et al.* Global hotspots and correlates of emerging zoonotic diseases. *Nat. Commun.* **8**, 1124 (2017).
 54. Friedman, J. H. Greedy Function Approximation: A Gradient Boosting Machine. *The Annals of Statistics* vol. 29 1189–1232.
 55. Natekin, A. & Knoll, A. Gradient boosting machines, a tutorial. *Front. Neurobot.* **7**, 21 (2013).
 56. Hay, S. I. *et al.* Global mapping of infectious disease. *Philos. Trans. R. Soc. B Biol. Sci.* **368**, 20120250–20120250 (2013).
 57. Svetnik, V. *et al.* Random Forest: A Classification and Regression Tool for Compound Classification and QSAR Modeling. *J. Chem. Inf. Comput. Sci.* **43**, 1947–1958 (2003).
 58. Chen, T. & Guestrin, C. XGBoost: A scalable tree boosting system. in *Proceedings of the ACM SIGKDD International Conference on Knowledge Discovery and Data Mining* vols 13-17-August-2016 785–794 (Association for Computing Machinery, 2016).
 59. Babayan, S. A., Orton, R. J. & Streicker, D. G. Predicting reservoir hosts and arthropod vectors from evolutionary signatures in RNA virus genomes. *Science (80-.).* **362**, 577–580 (2018).
 60. Dallas, T., Park, A. W. & Drake, J. M. Predicting cryptic links in host-parasite networks. *PLOS Comput. Biol.* **13**, e1005557 (2017).
 61. Lobo, J. M., Jiménez-Valverde, A. & Real, R. AUC: a misleading measure of the performance of predictive distribution models. *Glob. Ecol. Biogeogr.* **17**, 145–151 (2008).
 62. Barbet-Massin, M., Jiguet, F., Albert, C. H. & Thuiller, W. Selecting pseudo-absences for species distribution models: how, where and how many? *Methods Ecol. Evol.* **3**, 327–338 (2012).